# Propagating Uncertainty in Reinforcement Learning via Wasserstein Barycenters

**Alberto Maria Metelli**[*]
DEIB
Politecnico di Milano
Milan, Italy
albertomaria.metelli@polimi.it

**Amarildo Likmeta**[*]
DEIB
Politecnico di Milano
Milan, Italy
amarildo.likmeta@polimi.it

**Marcello Restelli**
DEIB
Politecnico di Milano
Milan, Italy
marcello.restelli@polimi.it

## Abstract

How does the uncertainty of the value function propagate when performing temporal difference learning? In this paper, we address this question by proposing a Bayesian framework in which we employ approximate posterior distributions to model the uncertainty of the value function and Wasserstein barycenters to propagate it across state-action pairs. Leveraging on these tools, we present an algorithm, *Wasserstein Q-Learning* (WQL), starting in the tabular case and then, we show how it can be extended to deal with continuous domains. Furthermore, we prove that, under mild assumptions, a slight variation of WQL enjoys desirable theoretical properties in the tabular setting. Finally, we present an experimental campaign to show the effectiveness of WQL on finite problems, compared to several RL algorithms, some of which are specifically designed for exploration, along with some preliminary results on Atari games.

## 1 Introduction

Effectively balancing exploration and exploitation is a key challenge in Reinforcement Learning [RL, 43]. When an agent takes decisions under uncertainty, it faces the dilemma between *exploiting* the information collected so far to execute what is believed to be the best action or to choose a possibly suboptimal action to *explore* new portions of the environment and gather new information, leading to more profitable behaviors in the future. Traditional exploration strategies, such as $\epsilon$-greedy and Boltzmann exploration [43], inject random noise into the action-selection process, i.e., the policy, to guarantee that each action is tried often enough. Although these methods allow RL algorithms to learn the optimal value function under mild assumptions [39], they are not *efficient*, since exploration is random and not driven by confidence on the value function estimate. Therefore, they might converge towards the optimal behavior after an exponential number of steps [24].

The exploration-exploitation dilemma has been extensively analyzed in the RL community, focusing on the definition of proper indices for *provably-efficient* exploration and devising algorithms with strong theoretical guarantees [25, 11, 21, 30]. Most of these algorithms are inherently model-based, i.e., they need to maintain and update estimates of the environment dynamics and the reward function during the learning process. For this reason, model-based methods are rather unsuited to problems with large state spaces and inapplicable to continuous environments. Apart from rare exceptions [41], the RL community has only recently focused on devising efficient model-free exploration strategies. Some works have succeeded in obtaining provably-efficient algorithms [35, 31, 23]; whereas others are more empirically-oriented [30, 29, 6].

---

[*]Equal contribution.

A fundamental step towards efficient exploration is the quantification of the *uncertainty* of the value function. The notion of uncertainty is formalized in Bayesian statistics by means of a *posterior* distribution. Bayesian Reinforcement Learning incorporates the Bayesian inference tools to provide a principled way to address the exploration-exploitation dilemma [20]. However, these methods rarely exploit the specific way in which the uncertainty propagates through the Bellman equation. Recently, in [28] a partial answer has been provided, proposing an uncertainty Bellman inequality; although no posterior distribution is explicitly considered.

In this paper, we propose a novel Bayesian framework to address the problem of exploration using posterior distributions over the value function. Specifically, we focus on how to *model* and *propagate* uncertainty when performing temporal-difference learning (Section 3). Moreover, we show how to use this uncertainty information to effectively explore the environment. Finally, we combine these elements to build our algorithm: *Wasserstein Q-Learning* (WQL, Section 4). Similarly to Bayesian Q-Learning [15], we equip each state-action pair with an approximate posterior distribution (named *Q-posterior*), whose goal is to quantify the uncertainty of the value function. Whenever a transition occurs, we update our distribution, in a temporal difference [TD, 43] fashion, in order to incorporate all sources of uncertainty: i) the one due to the sample estimate of the reward function and environment dynamics; ii) the uncertainty injected using the estimate of the next-state value function. Rather than employing a standard Bayesian update, we resort to a variational approach to approximate the posterior distribution, based on Wasserstein barycenters [2]. Recently, several works have embedded into RL algorithms notions coming from *Optimal Transport* [OT, 51], especially the Wasserstein metric, to improve the learning abilities of policy search algorithms [34] or in the filed of robust RL [1]. Furthermore, we prove in Section 5, that a slight modification of WQL, in tabular domains, is *PAC-MDP in the average loss setting* [42]. After examining the related literature (Section 6), we present an experimental evaluation on tabular environments to show the effectiveness of WQL, compared to the classic RL algorithms, some of which specifically designed for exploration (Section 7.1). Finally, we provide some preliminary results on the application of WQL to deep architectures (Section 7.2). The proofs of all results are reported in Appendix B. The implementation of the proposed algorithms can be found at `https://github.com/albertometelli/wql`.

## 2 Preliminaries

In this section, we provide the notation and the basic notions we will use in the following. Given a set $\mathcal{X}$, we denote with $\mathscr{P}(\mathcal{X})$ the set of all probability measures over $\mathcal{X}$.

**Markov Decision Processes** A discrete-time Markov Decision Process [MDP, 36] is defined as a 5-tuple $\mathcal{M} = (\mathcal{S}, \mathcal{A}, \mathcal{P}, \mathcal{R}, \gamma)$, where $\mathcal{S}$ is the state space, $\mathcal{A}$ is the (finite) action space, $\mathcal{P} : \mathcal{S} \times \mathcal{A} \to \mathscr{P}(\mathcal{S})$ is a Markovian transition model, $\mathcal{R} : \mathcal{S} \times \mathcal{A} \to \mathscr{P}(\mathbb{R})$ is a Markovian reward model, $\gamma \in [0, 1)$ is the discount factor. The behavior of an agent is defined by means of a Markovian policy $\pi : \mathcal{S} \to \mathscr{P}(\mathcal{A})$. Whenever the environment is in state $s \in \mathcal{S}$, the agent performs action $A \sim \pi(\cdot|s)$ and the environment transitions to the next state $S' \sim \mathcal{P}(\cdot|s, A)$ providing the agent with the reward $R \sim \mathcal{R}(\cdot|s, A)$. We assume $|R| \leq r_{\max} < +\infty$ almost surely. We indicate with $r(s, a) = \mathbb{E}_{R \sim \mathcal{R}(\cdot|s,a)}[R]$ the expected reward obtained by taking action $a \in \mathcal{A}$ in state $s \in \mathcal{S}$. Given a policy $\pi$ we define the state-value function, or V-function, as $v_\pi(s) = \mathbb{E}_{A \sim \pi(\cdot|s), S' \sim \mathcal{P}(\cdot|s,A)}[r(s, A) + \gamma v_\pi(S')]$. The action-value function, or Q-function, is given by $q_\pi(s, a) = r(s, a) + \gamma \mathbb{E}_{S' \sim \mathcal{P}(\cdot|s,a), A' \sim \pi(\cdot|S')}[q_\pi(S', A')]$. The optimal action-value function is defined as $q^*(s, a) = \sup_{\pi \in \Pi}\{q_\pi(s, a)\}$ for all $(s, a) \in \mathcal{S} \times \mathcal{A}$ and it satisfies the optimal Bellman equation: $q^*(s, a) = r(s, a) + \gamma \mathbb{E}_{S' \sim \mathcal{P}(\cdot|s,a)}[\max_{a' \in \mathcal{A}}\{q^*(S', a')\}]$. The boundedness of the reward function implies that the Q-function is uniformly bounded, i.e., $|q^*(s, a)| \leq q_{\max} \leq r_{\max}/(1 - \gamma)$. Then, an optimal policy $\pi^*$ is any policy that plays only greedy actions w.r.t. $q^*$, i.e., for all $s \in \mathcal{S}$ we have $\pi^*(\cdot|s) \in \mathscr{P}(\arg\max_{a \in \mathcal{A}}\{q^*(s, a)\})$.

**Temporal Difference Learning** Temporal-difference methods update the estimate of the optimal Q-function using the estimates of the next states V-functions [43]. For TD(0), we have that whenever a $(S_t, A_t, S_{t+1}, R_{t+1})$ tuple is collected, the *temporal difference update rule* is executed:

$$q_{t+1}(S_t, A_t) = (1 - \alpha_t)\, q_t(S_t, A_t) + \alpha_t\, (R_{t+1} + \gamma v_t(S_{t+1})), \qquad (1)$$

where $q_t$ is the estimated Q-function at time $t$, $\alpha_t \geq 0$ is a learning rate, and $v_t$ is an estimate of the V-function at time $t$. Different choices for $v_t$ generate different learning algorithms. If $v_t(S_{t+1}) =$

$q_t(S_{t+1}, A_{t+1})$ we get the SARSA update [38], if $v_t(S_{t+1}) = \mathbb{E}_{A \sim \pi_t(\cdot|S_{t+1})}[q_t(S_{t+1}, A)]$ we get the Expected SARSA update [50], being $\pi_t$ the exploration policy played at time $t$, and if $v_t(S_{t+1}) = \max_{a \in \mathcal{A}}\{q_t(S_{t+1}, a)\}$ we are performing Q-learning [52].

**Wasserstein Barycenters**   Let $(\mathcal{X}, d)$ be a complete separable metric (Polish) space and $x_0 \in \mathcal{X}$ be an arbitrary point. For each $p \in [1, +\infty)$ we define $\mathscr{P}_p(\mathcal{X})$ as the set of all probability measures $\mu$ over $(\mathcal{X}, \mathscr{F})$ such that $\mathbb{E}_{X \sim \mu}[d(X, x_0)^p] < +\infty$. Let $\mu, \nu \in \mathscr{P}_p(\mathcal{X})$, the $L^p$-Wasserstein distance between $\mu$ and $\nu$ is defined as [51]:

$$W_p(\mu, \nu) = \left( \inf_{\rho \in \Gamma(\mu, \nu)} \mathbb{E}_{X, Y \sim \rho} \left[ d(X, Y)^p \right] \right)^{1/p}, \tag{2}$$

where $\Gamma(\mu, \nu)$ is the set of all probability measures on $\mathcal{X} \times \mathcal{X}$ (couplings) with marginals $\mu$ and $\nu$. With little abuse of notation, we will indicate with $W_p(X, Y) = W_p(\mu, \nu)$, whenever clear from the context. The Wasserstein distance comes from the optimal transport community. Intuitively, it represents the "cost" to move the probability mass to turn one distribution into the other. Given a set of probability measures $\{\nu_i\}_{i=1}^n$, belonging to the class $\mathcal{N}$, and a set of weights $\{\xi_i\}_{i=1}^n$, $\sum_{i=1}^n \xi_i = 1$ and $\xi_i \geq 0$, the $L^2$-Wasserstein barycenter is defined as [2]:

$$\bar{\nu} = \arg\inf_{\nu \in \mathcal{N}} \left\{ \sum_{i=1}^n \xi_i W_2(\nu, \nu_i)^2 \right\}. \tag{3}$$

## 3   How to Model and Propagate Uncertainty?

In this section, we introduce a unifying Bayesian framework for exploration in RL that employs (approximate) posterior distributions to *model* uncertainty of value functions (Section 3.1) and Wasserstein barycenters to *propagate* uncertainty when performing TD updates (Section 3.2). Furthermore, we discuss how to leverage on the Q-posteriors to estimate the action that attains the maximum return in each state (Section 3.3) and to effectively explore the environment (Section 3.4).

### 3.1   Modeling Uncertainty via Q-Posteriors

Taking inspiration from Bayesian approaches to RL [15, 20], for each state $s \in \mathcal{S}$ and action $a \in \mathcal{A}$ we maintain a probability distribution $\mathcal{Q}(s, a)$, which we call *Q-posterior*, representing a (possibly approximate) posterior distribution of the Q-function estimate. This distribution will depend on the underlying MDP, in particular, the environment dynamics $\mathcal{P}$ and reward model $\mathcal{R}$, and on the updates of the Q-function estimates performed. As in a model-free scenario we cannot represent such distribution exactly, we employ a class of approximating probability distributions $\mathcal{Q} \subseteq \mathscr{P}(\mathbb{R})$. Similarly to usual value functions, we introduce the *V-posterior* $\mathcal{V}(s)$ which represents the (possibly approximate) posterior distribution of V-function, that combines the uncertainties modeled by the Q-posteriors $\mathcal{Q}(s, a)$. Furthermore, being the V-function defined, in the usual framework, as the expectation of the Q-function over the action space, i.e., $v_\pi(s) = \mathbb{E}_{A \sim \pi}[q_\pi(s, a)]$, it is natural to define, in our setting, the V-posterior $\mathcal{V}(s)$ as the Wasserstein barycenter of the Q-posteriors $\mathcal{Q}(s, a)$.[2]

**Definition 3.1** (V-posterior). *Given a policy $\bar{\pi}$ and a state $s \in \mathcal{S}$, we define the V-posterior $\mathcal{V}(s)$ induced by the Q-posteriors $\mathcal{Q}(s, a)$ with $a \in \mathcal{A}$ as the Wasserstein barycenter of the $\mathcal{Q}(s, a)$:*

$$\mathcal{V}(s) \in \arg\inf_{\mathcal{V} \in \mathcal{Q}} \left\{ \mathbb{E}_{A \sim \bar{\pi}(\cdot|s)} \left[ W_2 \left( \mathcal{V}, \mathcal{Q}(s, A) \right)^2 \right] \right\}. \tag{4}$$

When the policy $\bar{\pi}$ is known, the expectation over the action space can be computed as we are assuming that $\mathcal{A}$ is finite. In a prediction problem, policy $\bar{\pi}$ is a fixed policy, whereas, in a control problem, $\bar{\pi}$ is a policy aimed at properly selecting the best action in state $s$ accounting for the uncertainty modeled by the Q-posterior (see Section 3.3). Moreover, when $\mathcal{Q}(s, a)$ are deterministic distributions, $\mathcal{V}(s)$ is a deterministic distribution too centered in the mean of the $\mathcal{Q}(s, a)$. In this way, we obtain the usual V-function definition (see Proposition A.3).

It is important to stress that our approach is rather different from *Distributional Reinforcement Learning* [9, 13, 12, 37]. Indeed, we employ a distribution to represent the *uncertainty* of the Q-function estimate and not the *intrinsic randomness* of the return. The two distributions are clearly related and both depend on the stochasticity of the reward and of the transition model. However, in our approach the stochasticity refers to the uncertainty on the Q-function estimate which reduces as the number of updates increases, being a sample mean.[3]

## 3.2 Propagating Uncertainty via Wasserstein Barycenters

In this section, we discuss the problem of uncertainty propagation, i.e., how to deal with the update of the Q-posteriors when experiencing a transition $(S_t, A_t, S_{t+1}, R_{t+1})$. Whenever a TD update (Equation (1)) is performed, there are two sources of uncertainty involved. First, we implicitly estimate the environment dynamics $\mathcal{P}(\cdot|S_t, A_t)$ and the reward model $\mathcal{R}(\cdot|S_t, A_t)$ using a set of sampled transitions $(S_t, A_t, S_{t+1}, R_{t+1})$. Second, when using the V-function estimates of the next states $v_t(S_{t+1})$ we bring into $q_{t+1}(S_t, A_t)$ part of the uncertainty of $v_t(S_{t+1})$ and they become correlated. For this reason, the standard Bayesian posterior update, used for instance in Bayesian Q-learning [15], becomes rather inappropriate as it assumes that the samples are independent, which is clearly not true. We argue that, rather than using a Bayesian update, when we have a Q-posterior $\mathcal{Q}_t(S_t, A_t)$ and a V-posterior $\mathcal{V}_t(S_{t+1})$ we can combine them using a notion of barycenter, which does not require the independence assumption. We formalize this idea in the following update rule.

**Definition 3.2** (Wasserstein Temporal Difference). *Let $\mathcal{Q}_t$ be the current Q-posterior, given a transition $(S_t, A_t, S_{t+1}, R_{t+1})$, we define the* TD-target-posterior *as $\mathcal{T}_t = R_{t+1} + \gamma \mathcal{V}_t(S_{t+1})$. Let $\alpha_t \geq 0$ be the learning rate, we define the* Wasserstein Temporal Difference *(WTD) update rule as:*

$$\mathcal{Q}_{t+1}(S_t, A_t) \in \underset{\mathcal{Q} \in \mathscr{Q}}{\arg\inf} \left\{ (1 - \alpha_t) W_2 \left( \mathcal{Q}, \mathcal{Q}_t(S_t, A_t) \right)^2 + \alpha_t W_2 \left( \mathcal{Q}, \mathcal{T}_t \right)^2 \right\}. \tag{5}$$

Therefore, the new Q-posterior $\mathcal{Q}_{t+1}(S_t, A_t)$ is the Wasserstein barycenter between the current Q-posterior $\mathcal{Q}_t(S_t, A_t)$ and the TD-target posterior $\mathcal{T}_t = R_{t+1} + \gamma \mathcal{V}_t(S_{t+1})$, which in turn embeds information of the current transition (i.e., the reward $R_{t+1}$ and the next state $S_{t+1}$) and the next-state V-posterior $\mathcal{V}_t(S_{t+1})$. It is worth noting that the two terms appearing in Equation (5) account for all sources of uncertainty. Indeed, the first term $W_2 \left( \mathcal{Q}, \mathcal{Q}_t(S_t, A_t) \right)$ avoids moving too far from the current estimation $\mathcal{Q}_t(S_t, A_t)$, as we are performing the update experimenting a single transition, whereas $W_2 \left( \mathcal{Q}, \mathcal{T}_t \right)$ allows bringing in the new Q-posterior the V-posterior of the next-state $\mathcal{V}_t(S_{t+1})$ (including its uncertainty). We stress the analogy with the standard TD update in the following result.

**Proposition 3.1.** *If $\mathscr{Q}$ is the set of deterministic distributions over $\mathbb{R}$, then the WTD update rule (Equation (5)) has a unique solution that corresponds to the TD update rule (Equation (1)).*

Supporting deterministic distributions, as the Q-posteriors, is fundamental for our method that models a sample mean, whose variance reduces as the number of samples increases, moving towards a deterministic distribution. This justifies the choice of the Wasserstein metric over other distributional distances (e.g., $\alpha$-divergences). The choice of the prior for $\mathcal{Q}_0$ plays an important role, along with the learning rate schedule $\alpha_t$. We will show in Section 5 that specific choices of $\mathcal{Q}_0$ and $\alpha_t$, for a particular class of distributions $\mathscr{Q}$, allow achieving PAC-MDP property in the average loss setting.

## 3.3 Estimating the Maximum Expected Value

The TD-target-posterior $\mathcal{T}_t = R_{t+1} + \gamma \mathcal{V}_t(S_{t+1})$ is defined in terms of the next state V-posterior $\mathcal{V}_t(S_{t+1})$. In a control problem, we aim at learning the optimal Q-function $q^*$ and, thus, we are interested in propagating back to $\mathcal{Q}_{t+1}(S_t, A_t)$ a V-posterior $\mathcal{V}_t(S_{t+1})$ related to the optimal action to be taken in the next state.[4] This can be performed by a suitable choice of the policy $\overline{\pi}$, as in Definition 3.1. A straightforward approach consists in propagating the Q-posterior $\mathcal{Q}(S_{t+1}, a)$ of the action with the highest estimated mean, i.e., $\overline{\pi}^M(\cdot|s) \in \mathscr{P}\left(\arg\max_{a \in \mathcal{A}}\{\mathbb{E}_{Q \sim \mathcal{Q}(s,a)}[Q]\}\right)$.

Table 1: Probability density function (pdf), Wasserstein Temporal Difference (WTD) update rule and computation of the V-posterior for Gaussian and Particle posterior distributions.

| $\mathscr{Q}$ | pdf | WTD and V-posterior |
|---|---|---|
| Gaussian | $\dfrac{\exp\left\{-\frac{1}{2}\left(\frac{x-m(s,a)}{\sigma(s,a)}\right)^2\right\}}{\sqrt{2\pi\sigma^2(s,a)}}$ | $m_{t+1}(S_t, A_t) = \alpha_t m_t(S_t, A_t) + (1-\alpha_t)(R_{t+1} + \gamma m_t(S_{t+1}))$ $\sigma_{t+1}(S_t, A_t) = \alpha_t \sigma_t(S_t, A_t) + (1-\alpha_t)\gamma\sigma_t(S_{t+1})$ $m(s) = \mathbb{E}_{A \sim \overline{\pi}(\cdot|s)}[m(s, A)]$ $\sigma(s) = \mathbb{E}_{A \sim \overline{\pi}(\cdot|s)}[\sigma(s, A)]$ |
| Particle | $\sum_{j=1}^M w_j \delta(x - x_j(s,a))$ $x_1(s,a) \leq ... \leq x_M(s,a)$ $\sum_{i=j}^M w_j = 1$ and $w_j \geq 0$ | $x_{j,t+1}(S_t, A_t) = \alpha_t x_{j,t}(S_t, A_t) + (1-\alpha_t)(R_{t+1} + \gamma x_{j,t}(S_{t+1}))$ $x_j(s) = \mathbb{E}_{A \sim \overline{\pi}(\cdot|s)}[x_j(s, A)], \ j = 1, 2, ..., M$ |

We refer to this approach as *Mean Estimator* (ME) for the maximum. However, when posterior distributions are available, we can use them to define a wiser way to estimate the V-posterior of the next state.[5] A first method based on Optimism in the Face of Uncertainty [OFU, 3] consists in selecting the action that maximizes a statistical upper bound $u^\delta(s,a)$ of the Q-posterior, i.e., $\overline{\pi}^O(\cdot|s) \in \mathscr{P}\left(\arg\max_{a \in \mathcal{A}}\{u^\delta(s,a)\}\right)$. We will refer to this method as *Optimistic Estimator* (OE). However, if we want to make full usage of the Q-posteriors, we can resort to the *Posterior Estimator* (PE) of the maximum, based on Posterior Sampling [PS, 47]. In this case, each action contributes to the update rule weighted by the probability of being the optimal action, i.e., $\overline{\pi}^P(a|s) = \Pr_{Q_{s,a} \sim \mathcal{Q}(s,a)}(a \in \arg\max_{a' \in \mathcal{A}}\{Q_{s,a'}\})$.

### 3.4 Exploring using the Q-posteriors

In the previous section, we have introduced two approaches that exploit the Q-posterior to properly define the V-posterior of the next state, using specific policies $\overline{\pi}$. These policies can also be used to implement effective exploration strategies aware of the uncertainty. Using the optimistic policy $\overline{\pi}^O$ in each state, we play (deterministically) the action that maximizes the statistical upper bound on the estimated Q-function $u^\delta(s,a)$, we call this strategy *Optimistic Exploration* (OX). Instead, we can directly use the posterior policy $\overline{\pi}^P$ to sample the action from the Q-posterior $\mathcal{Q}(s,a)$. Thus, in *Posterior Exploration* (PX), each action is played with the probability of being optimal.

## 4 Wasserstein Q-Learning

The ideas presented so far can be combined in an algorithm, *Wasserstein Q-Learning* (WQL), whose pseudocode is reported in Algorithm 1.

We developed our approach for a generic class of distributions $\mathscr{Q}$, however, in practice, we focus on two specific classes: *Gaussian* posteriors (G-WQL) and *Particle* posteriors (P-WQL), i.e., a mixture of $M > 1$ Dirac deltas. For both classes the Wasserstein Barycenter is unique and can be computed in closed form (see Appendix A.3).[6] In Table 1, we summarize the main relevant features of these distributions classes. WQL simply needs

---

**Input**: a prior distribution $\mathcal{Q}_0$, a step size schedule $(\alpha_t)_{t \geq 0}$, an exploration policy schedule $(\pi_t)_{t \geq 0}$
1: Initialize $\mathcal{Q}(s,a)$ with the prior $\mathcal{Q}_0$
2: **for** $t = 1, 2, ...$ **do**
3:     Take action $A_t \sim \pi_t(\cdot|S_t)$
4:     Observe $S_{t+1}$ and $R_{t+1}$
5:     Compute $\mathcal{V}_t(S_{t+1})$ using Equation (4)
6:     Update $\mathcal{Q}_{t+1}(S_t, A_t)$ using Equation (5)
7: **end for**

Algorithm 1: Wasserstein Q-Learning.

---

to store the parameters of the Q-posterior for every state-action pair ($m(s,a)$ and $\sigma(s,a)$ for G-WQL and $x_j(s,a)$ for P-WQL). Therefore, unlike the majority of provably-efficient algorithms, it can be extended straightforwardly to continuous state spaces as long as we adopt a function approximator for the parameters of the posterior. For instance, we could approximate $m(s,a)$ and $\sigma(s,a)$ or the particles $x_j(s,a)$ using a neural network with multiple heads. For this reason, our method easily

applies to deep architecture by adopting a network that directly outputs the posterior parameters, instead of the value function (see Section 7.2).

# 5 Theoretical Analysis

In this section, we show that WQL, with some modifications, enjoys desirable theoretical properties in the tabular setting. We start providing a modification of the WTD update rule that will be used for the analysis; then we prove that with such modification our algorithm, under certain assumptions, is PAC-MDP in the average loss setting [42].

**Definition 5.1** (Modified Wasserstein Temporal Difference). *Let $\mathcal{Q}_t$ be the current Q-posterior and $\mathcal{Q}_b$ be a zero-mean distribution, given a transition $(S_t, A_t, S_{t+1}, R_{t+1})$, we define the* TD-target-posterior *as $\mathcal{T}_t = R_{t+1} + \gamma \mathcal{V}_t(S_{t+1})$. Let $\alpha_t, \beta_t \geq 0$ be the learning rates, we define the* Modified Wasserstein Temporal Difference *(MWTD) update rule as:*

$$\widetilde{\mathcal{Q}}_{t+1}(S_t, A_t) \in \underset{\mathcal{Q} \in \mathscr{Q}}{\arg\inf} \left\{ (1 - \alpha_t) W_2 \left( \mathcal{Q}, \widetilde{\mathcal{Q}}_t(S_t, A_t) \right)^2 + \alpha_t W_2 \left( \mathcal{Q}, \mathcal{T}_t \right)^2 \right\},$$

$$\mathcal{Q}_{t+1}(S_t, A_t) \in \underset{\mathcal{Q} \in \mathscr{Q}}{\arg\inf} \left\{ W_2 \left( \mathcal{Q}, \widetilde{\mathcal{Q}}_{t+1}(S_t, A_t) + \beta_t \mathcal{Q}_b \right)^2 \right\},$$

We will denote the algorithm employing this update rule as *Modified Wasserstein Q-Learning* (MWQL). The reason why we need to change the WTD lies in the fact that the uncertainty on the Q-function value (the Q-posterior) is, as already mentioned, the contribution of two terms: i) the uncertainty on the reward and transition model; ii) the uncertainty on the next-state Q-function. These terms need to be averaged into the Q-posterior at different speeds. If $n_t(s, a)$ is the number of times $(s, a)$ is visited up to time $t$, (i) has to reduce proportionally to $1/\sqrt{n_t(s, a)}$ being a sample mean, while (ii) is averaged with coefficients proportional to $1/n_t(s, a)$. Therefore, we should keep the two sources of uncertainty separated. To this end, we use an additional distribution $\mathcal{Q}_b$ to prevent the uncertainty from reducing too fast.

The notion of *PAC-MDP in the average loss setting* [42] is a relaxation of the classical PAC-MDP notion introduced in [24], in which we consider the actual reward received by the algorithm while learning, instead of the expected values over future policies. We recall the definitions given in [42].

**Definition 5.2** (Definition 4 of [42]). *Suppose a learning algorithm $\mathfrak{A}$ is run for $T$ steps. Consider partial sequence $S_0, R_1, ..., S_{T-1}, R_T, S_T$ visited by $\mathfrak{A}$. The* instantaneous loss *of the agent at time $t$ is $il_{\mathfrak{A}}(t) = v^*(S_t) - \sum_{i=t}^T \gamma^{i-t} R_{i+1}$. The quantity $\mathcal{L}_{\mathfrak{A}} = \frac{1}{T} \sum_{t=1}^T il_{\mathfrak{A}}(t)$ is called the* average loss.

Then, a learning algorithm $\mathfrak{A}$ is PAC-MDP in the average loss setting if for any $\epsilon \geq 0$ and $\delta \in [0, 1]$, we can choose a value $T$, polynomial in the relevant quantities $(1/\epsilon, 1/\delta, |\mathcal{S}|, |\mathcal{A}|, 1/(1 - \gamma))$, such that the average loss $\mathcal{L}_{\mathfrak{A}}$ of the agent (following the learning algorithm $\mathfrak{A}$) on a trial of $T$ steps is guaranteed to be less than $\epsilon$ with probability at least $1 - \delta$.

In the following, we will restrict our attention to MWQL with Gaussian posterior, optimistic estimator (OE) and optimistic exploration policy (OX). We leave the analysis of the posterior sampling exploration (PX) as future work. To prove the main result we need an intermediate result.

**Theorem 5.1.** *Let $S_0, ..., S_{T-1}, S_T$ be the sequence of states and actions visited by MWQL with Gaussian posterior, OE and OX. Then, there exists a prior $\mathcal{Q}_0$ and a zero-mean distribution $\mathcal{Q}_b$ and a learning rate schedule for $(\alpha_t, \beta_t)_{t \geq 0}$ (whose values are reported in Appendix B.1), such that for any $\delta \in [0, 1]$, with probability at least $1 - \delta$ it holds that:[7]*

$$\sum_{t=1}^T [v^*(S_t) - v_{\mathfrak{A}}(S_t)] \leq \mathcal{O} \left( \frac{q_{\max}}{(1 - \gamma)^{\frac{3}{2}}} \sqrt{|\mathcal{S}||\mathcal{A}|T \log \frac{|\mathcal{S}||\mathcal{A}|T}{\delta}} \right), \tag{6}$$

*where $v_{\mathfrak{A}}$ is the value function induced by the (non-stationary) policy played by algorithm $\mathfrak{A}$.*

From this result, we can exploit an analysis similar to [42] to prove that MWQL with Gaussian posterior, OE and OX is PAC-MDP in the average loss setting.

**Theorem 5.2.** *Under the hypothesis of Theorem 5.1, MWQL with Gaussian posterior, OE and OX is PAC-MDP in the average loss setting, i.e., for any $\epsilon \geq 0$ and $\delta \in [0, 1]$, after*

$$T = \mathcal{O}\left(\frac{q_{\max}^2|\mathcal{S}||\mathcal{A}|}{\epsilon^2(1-\gamma)^3} \log \frac{q_{\max}^2|\mathcal{S}|^2|\mathcal{A}|^2}{\delta\epsilon^2(1-\gamma)^3}\right)$$

*steps we have that the average loss $\mathcal{L}_{\mathfrak{A}} \leq \epsilon$ with probability at least $1 - \delta$.*

The per-step computational complexity of MWQL is $\mathcal{O}(\log|\mathcal{A}|)$ as we can maintain the upper bounds of the Q-function as a max-priority queue [40] and the space complexity is $\mathcal{O}(|\mathcal{S}||\mathcal{A}|)$.

Despite the theoretical guarantees, MWQL turns out to be often impractical for two main reasons. First, MWQL cannot be extended to continuous MDPs, as $\alpha_t$ and $\beta_t$ are defined in terms of number of visits $n(s, a)$ (Equation (20)), which can only be computed for finite MDPs. Second, as many provably efficient RL algorithms, MWQL is extremely conservative, leading to very slow convergence. This is why most provably efficient RL algorithms, when used in practice, are run with non-theoretical values of hyperparameters. In this sense, WQL can be seen as a "practical" version of MWQL in which $\alpha_t$ is treated as a normal hyper-parameter and $\beta_t = 0$.

## 6   Related Works

A variety of approaches has been proposed in the RL literature to tackle the exploration-exploitation trade-off [44]. We consider only those that do not assume the availability of a simulator of the environment [26]. A first dimension of classification is the RL setting they consider: *finite-horizon*, *discounted* or *undiscounted*. Finite-horizon MDPs are a convenient framework to devise provably-efficient exploration algorithms with theoretical guarantees on the *regret* [32, 14, 5]. Recently, in [23] it was shown that Q-learning, in the finite-horizon setting, can be made efficient by resorting to suitable exploration bonuses. Similar results have been proposed in the infinite-horizon undiscounted case. The main challenge of this class of problems is the connection structure of the MDP [7]. Early approaches [25, 4, 46, 21] impose restrictive requirements on either mixing/hitting times or diameter, which have been progressively relaxed [19]. A significant part of the early provably-efficient algorithms considers the discounted setting [25, 11, 41, 45, 27]. However, their theoretical guarantees are based on the notion of PAC-MDP [24] rather than on regret.

Another relevant dimension is the kind of policy used for exploration. Taking inspiration from the Multi Armed Bandit [MAB, 10] framework, two main approaches have been proposed: Optimism in the Face of Uncertainty [3] and Thompson Sampling [47]. Most exploration algorithms employ the optimistic technique, selecting actions from the optimal policy of an optimistic approximation of the MDP [21] or of the value function directly [41, 23]. Some methods, instead, use a posterior sampling approach in which either the entire MDP or a value function is sampled from a (possibly approximate) posterior distribution.

Inspired by these methods, numerous practical variants have been devised. Exploration bonuses, based on pseudo-counts [8, 33], mimicking optimism, have been applied with positive results to deep architectures. Likewise, with the idea of approximating a posterior distribution, *Bootstrapped DQN* [30] and *Bayesian DQN* [6] succeeded in solving challenging Atari games. Recently, new results of sample-efficiency beyond tabular domains have been derived [22].

## 7   Experiments

In this section, we provide an experimental evaluation of WQL on tabular domains along with some preliminary results on Atari games (implementation details are reported in Appendix C).

### 7.1   Tabular Domains

We evaluate WQL on a set of RL tasks designed to emphasize exploration: the Taxi problem [15], the Chain [15], the River Swim [42], and the Six Arms [42]. We extensively test several WQL variants that differ on: i) the Q-posterior model (Gaussian G-WQL vs particle P-WQL); ii) the exploration strategy (optimistic OX vs posterior sampling PX), iii) the estimator of the maximum (ME, OE, and PE).

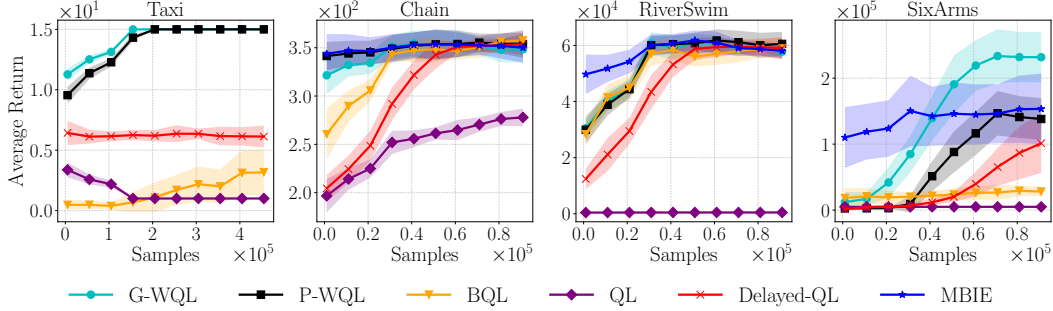

Figure 1: Online average return as a function of the number of samples, comparison of P-WQL and G-WQL with QL, BQL, Delayed-QL, and MBIE-EB. 10 runs, 95% c.i.

We compare these combinations with the classic Q-learning [QL, 52] (Boltzmann exploration), Bootstrapped Q-learning [BQL, 30] both with the double estimator [48], Delayed Q-learning [Delayed-QL, 41] and MBIE-EB [42].[8]

Figure 1 shows the *online* performance on the considered tabular tasks. While we tried all the WQL variants, due to space constraints, we show the best combination of exploration strategy and maximum estimator for both Gaussian and particle models (complete results are reported in Appendix C). We can see that WQL learns substantially faster than classical approaches, like QL, in tasks that require significant exploration, such as Taxi, Six Arms, or River Swim. Our algorithm also outperforms BQL in most tasks, except in the River Swim, where performances are not substantially different. Finally, we can see that across all the tasks WQL displays a faster learning curve w.r.t. to Delayed-QL. MBIE-EB outperforms WQL in small domains like Chain and RiverSwim, but not in SixArms. MBIE-EB was not

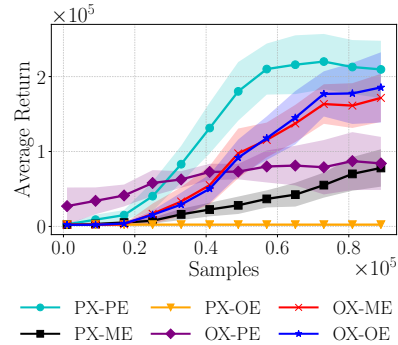

Figure 2: Online average return as a function of the number of samples for the different versions of G-WQL algorithm. 10 runs, 95% c.i.

tested on the Taxi domain as the number of states ($\sim 200$) makes the computational time demands prohibitive. We cross-validate the hyperparameter of Delayed Q-Learning and MBIE-EB.

Among the variants of WQL, we discovered that the choice of the exploration strategy and the maximum estimator are highly task dependent. However, we can see a general pattern across the tasks. As intuition suggests, being the exploration strategy and the maximum estimator closely related, the best combinations are: OX exploration with OE estimator and PX exploration with PE estimator. We illustrate in Figure 2 all the possible combinations of G-WQL on Six Arms, a domain in which exploration is essential. We can notice that the "hybrid" combinations, such as OX with PE and PX with OE are significantly outperformed by the more "coherent" ones.

## 7.2 Atari Games

We adapted WQL with the particle model to be used paired with deep architectures. For this purpose, similarly to Bootstrapped DQN [BDQN, 30], we use a network architecture with a head for each particle while the convolutional layers are shared among them. We compare the resulting algorithm, which we call *Particle DQN* (PDQN), with Double DQN [DDQN, 49], a classic benchmark in Deep-RL, and Bootstrapped DQN, specifically designed for deep exploration using Q-posteriors. To compare algorithms we consider *offline* scores, i.e., the scores collected using the current greedy policy. The goal of this experiment, conducted on three Atari games, is to prove that WQL, although designed to work in finite environments, can easily be extended to deep networks with potentially good results. In Figure 3, we can see that PDQN, compared to BDQN and DDQN, manages to

**Input**: a prior distribution $\{x_i\}_{i=1}^M$, a step size schedule $(\alpha_t)_{t \geq 0}$, an exploration policy schedule $(\pi_t)_{t \geq 0}$

1: Initialize a Q-function network with $M$ outputs $\{Q_j\}_{j=1}^M$ and parameters $\boldsymbol{\theta}$ and the target network with parameters $\boldsymbol{\theta}^- = \boldsymbol{\theta}$
2: **for** $t = 1, 2, ...$ **do**
3:     Take action $A_t \sim \pi_t(\cdot | S_t; \boldsymbol{\theta})$
4:     Store transition $(S_t, A_t, S_{t+1}, R_{t+1})$ in the replay buffer
5:     Sample random a batch of transitions $(S_l, A_l, S_{l+1}, R_{l+1})$ from the replay buffer
6:     Compute targets $y_j(S_{l+1}) = \mathbb{E}_{A \sim \overline{\pi}(\cdot | S_{l+1})}[Q_j(S_{l+1}, A; \boldsymbol{\theta}^-)]$ for each output $Q_j$ where $\overline{\pi} \in \{\overline{\pi}^M, \overline{\pi}^O, \overline{\pi}^P\}$ as in Section 3.3
7:     Perform a gradient descent step w.r.t. $\boldsymbol{\theta}$ on the objective $\sum_{j=1}^M (y_j(S_{t+1}) - Q_j(S_l, A_l; \boldsymbol{\theta}))^2$ and using the step size $\alpha_t$
8:     Periodically update target network $\boldsymbol{\theta}^- = \boldsymbol{\theta}$
9: **end for**

Algorithm 2: Particle DQN.

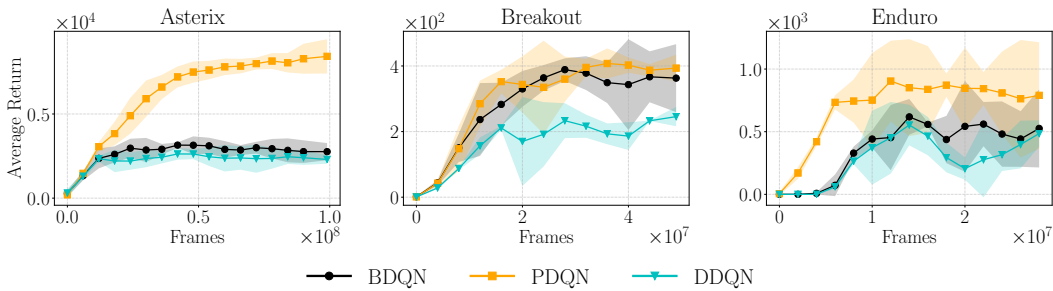

Figure 3: Offline average return of the greedy policy as a function of the number of collected frames, comparing PDQN, DDQN and BDQN on Asterix, Enduro and Breakout games. 5 runs, 95% c.i.

achieve higher scores in Asterix and Enduro, where exploration is needed, while achieving similar scores in Breakout. A relevant feature of PDQN is the particle initialization interval. Indeed, a narrower initial interval causes faster learning but might lead to premature convergence. In this sense, the initial interval becomes a hyperparameter of PDQN, which influences the amount of exploration and it is likely task-dependent. The pseudocode of PDQN is shown in Algorithm 2.

# 8 Discussion and Conclusions

In this paper, we presented a novel RL algorithm, Wasserstein Q-Learning (WQL), which addresses several issues related to efficient exploration in model-free RL. We discussed how to model uncertainty of the estimated Q-function by means of approximate posterior distributions (Q-posteriors). Then, we devised a variational method to propagate uncertainty across state-action pairs when performing TD learning, based on Wasserstein barycenters. The experimental evaluation allowed us to appreciate the properties of WQL. In tabular domains, whenever exploration is really necessary, our approach is able to significantly outperform TD methods even if designed specifically for exploration (e.g., Bootstrapped Q-Learning and Delayed Q-Learning). Although preliminary, the results on the Atari games are promising and need to be further investigated as future work in order to make WQL scale on complex environments. We believe that our algorithm contributes to bridging the gap between theory and practice of exploration in RL. WQL is a theoretically grounded method, equipped with guarantees in the average loss setting, but, at the same time, it is a very simple algorithm, easily extensible to deal with continuous domains.

## Footnotes

[2]The Wasserstein barycenter can be regarded as a way of averaging distributions [2].

[3] A notable difference w.r.t. the distributional RL is that the variance of our posterior distribution $\mathbb{V}\text{ar}_{Q \sim \mathcal{Q}(s,a)}[Q]$ vanishes as the number of updates grows to infinity.

[4] We stress that we are uninterested in modeling the distribution $\max_{a \in \mathcal{A}}\{\mathcal{Q}(s, a)\}$, but rather in exploiting the uncertainty modeled by $\mathcal{Q}(s, a)$ to properly perform the computation of the optimal action.

[5]This problem was treated in RL, without distributions, proposing several estimators, such as the double estimator [48] and the weighted estimator [17, 16].

[6]It is worth noting that, even for the Gaussian case, using the standard Bayesian posterior update is inappropriate, as the independence of the Q-function estimates across state-action pairs cannot be assumed.

[7]This performance index resembles the *regret* [21]. However, it is a weaker notion, being defined in terms of the trajectory generated by algorithm $\mathfrak{A}$, instead of the trajectories of an optimal policy.

[8]We are considering a discounted setting, thus, several provably efficient algorithms, like UCRL2 [21], PSRL [32], RLSVI [30], optimistic Q-learning [23] and UCBVI [5], cannot be compared as they consider either average reward or finite-horizon setting.

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
