[Supplementary Material]

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

[9]Different choices of exponent to which $d(x, Y)$ is raised generate different indexes of central tendency, like median for exponent 1 and mode for the exponent going to 0 in the limit.

[10]It can be easily proved that taking the value of $a$ that minimizes $\left(1 - \frac{a\gamma}{1-a}\right)^{-1}\sqrt{a}$ just changes the bound by a constant and does not modify the dependence on $(1-\gamma)$.

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

## Index of the Appendix

In the following, we briefly recap the contents of the Appendix.

– Appendix A provides additional details about the properties of the Wasserstein metric and the Wasserstein barycenters.

– Appendix B reports all proofs and derivations.

– Appendix C reports additional experiments in the tabular domain along with some implementation details.

## A   Details on Wasserstein distance

In this appendix, we provide some additional details on Wasserstein distance and some properties of our approximate posterior distribution models. It can be proved [51] that the functions $W_p$ are metrics on the sets $\mathscr{P}_p(\mathcal{X})$. Moreover, the following monotonicity property holds: $W_p(\mu, \nu) \leq W_q(\mu, \nu)$ if $p \leq q$. We will assume that all the involved probability measures $\mu$ admit cumulative distribution function (c.d.f.) $F_\mu$ and probability density function (p.d.f.) $f_\mu$ w.r.t. the Lebesgue measure. When $\mathcal{X} = \mathbb{R}$ and $d(x, y) = |x - y|$ is the Euclidean distance, the Wasserstein distance can be rephrased in terms of the quantile functions:

$$W_p(\mu, \nu) = \left( \int_0^1 \left| F_\mu^{-1}(t) - F_\nu^{-1}(t) \right|^p dt \right)^{1/p}, \tag{7}$$

where $F^{-1}$ is the quantile function, i.e., $F_\mu^{-1}(t) = \inf \{x \in \mathbb{R} : t \leq F_\mu(x)\}$. The $L^2$-Wasserstein distance admits a closed form for both the Gaussian and particle models. For two Gaussian distributions, considering just the univariate case, $\mu = \mathcal{N}(m_1, \sigma_1^2)$ and $\nu = \mathcal{N}(m_2, \sigma_2^2)$, we have [18]:

$$W_2(\mu, \nu)^2 = (m_1 - m_2)^2 + (\sigma_1 - \sigma_2)^2. \tag{8}$$

For the particle models, with the same weighting $w_j$ we have the following result.

**Proposition A.1.** *Let $\mu$ and $\nu$ be two mixture of $M$ Dirac deltas with the same weighting $w_j$ for $j = 1, 2, ..., M$ having $f_\mu(x) = \sum_{j=1}^M w_j \delta(x - x_j)$ and $f_\nu(x) = \sum_{j=1}^M w_j \delta(x - y_j)$ as p.d.f.s with $x_1 \leq x_2 \leq ... \leq x_M$ and $y_1 \leq y_2 \leq ... \leq y_M$. Then, the $L^2$-Wasserstein distance between $\mu$ and $\nu$ is given by:*

$$W_2(\mu, \nu)^2 = \sum_{j=1}^M w_j (x_j - y_j)^2. \tag{9}$$

*Proof.* We use Equation (7), thus we need to compute the quantile function of a mixture of $M$ Dirac deltas. Let us introduce the intervals $I_j = [t_{j-1}, t_j)$, $t_0 = 0$, $t_j = \sum_{k=1}^j w_k$ and $|I_j| = t_j - t_{j-1}$ for $j = 1, 2, ...N$. It is clear that:

$$F_\mu^{-1}(t) = \sum_{j=1}^M x_j \mathbb{1}_{I_j}(t), \quad F_\mu^{-1}(t) = \sum_{j=1}^M y_j \mathbb{1}_{I_j}(t),$$

where $\mathbb{1}_\mathcal{X}$ is the indicator function of the set $\mathcal{X}$. Thus we can compute the Wasserstein distance by employing Equation (7):

$$W_2(\mu, \nu)^2 = \int_0^1 \left| F_\mu^{-1}(t) - F_\nu^{-1}(t) \right|^2 dt$$

$$= \sum_{j=1}^M \int_{I_j} \left| F_\mu^{-1}(t) - F_\nu^{-1}(t) \right|^2 dt$$

$$= \sum_{j=1}^M \int_{I_j} (x_j - y_j)^2 dt$$

$$= \sum_{j=1}^M (x_j - y_j)^2 \int_{I_j} dt$$

$$= \sum_{j=1}^{M} w_j (x_j - y_j)^2,$$

where we observed that $\int_{I_j} \mathrm{d}t = t_j - t_{j-1} = w_j$. $\qquad \square$

## A.1 Approximation of an arbitrary distribution with a mixture of Deltas

In this section, we show that we are able to approximate an arbitrary distribution with a mixture of Deltas, provided that we consider a sufficiently large number of components.

**Proposition A.2.** *Let $\mu$ be an arbitrary probability measure over the interval $[a, b] \subset \mathbb{R}$ admitting $F_\mu$ as c.d.f. and let $\nu$ be a mixture of $M$ Dirac deltas $\widehat{x}_1 \leq \widehat{x}_2 \leq ... \leq \widehat{x}_M$ weighted by $w_j$ for $j = 1, 2, ..., M$ fixed. Then, the $L^2$-Wasserstein $W_2(\mu, \nu)$ has a unique minimizer:*

$$\widehat{x}_j = \frac{1}{|I_j|} \int_{I_j} F_\mu^{-1}(t) \mathrm{d}t, \quad j = 1, 2, \dots, M,$$

*where $I_j = [t_{j-1}, t_j)$, $t_0 = 0$, $t_j = \sum_{k=1}^{j} w_k$ and $|I_j| = t_j - t_{j-1}$ for $j = 1, 2, ...N$. In this case, the $L^2$-Wasserstain distance can be bounded as:*

$$W_2(\mu, \nu)^2 \leq \frac{(b-a)^2}{4} \max_{j=1,2,...,n} |I_j|.$$

*Proof.* Let us first compute the quantile function of $\nu$, i.e., $F_\nu^{-1}$:

$$F_\nu^{-1}(t) = \sum_{j=1}^{M} x_j \mathbb{1}_{I_j}(t). \tag{10}$$

Using Equation (7), the $L^2$-Wasserstein distance can be written as:

$$W_2(\mu, \nu)^2 = \int_0^1 \left( F_\mu^{-1}(t) - F_\nu^{-1}(t) \right)^2 \mathrm{d}t$$

$$= \sum_{j=1}^{M} \int_{I_j} \left( F_\mu^{-1}(t) - x_j \right)^2 \mathrm{d}t.$$

The objective is clearly convex, thus, we take the derivative w.r.t. $x_j$ and we get:

$$\frac{\partial W_2^2}{\partial x_j} = -2 \int_{I_j} \left( F_\mu^{-1}(t) - x_j \right) \mathrm{d}t = 0, \tag{11}$$

from which the first result follows, by observing that $\int_{I_j} \mathrm{d}t = |I_j|$. Let us now observe, since $F_\mu^{-1}$ is monotonically increasing, that for every $j$:

$$\int_{I_j} \left( F_\mu^{-1}(t) - \widehat{x}_j \right)^2 \mathrm{d}t \leq \int_{I_j} \left( F_\mu^{-1}(t) - \frac{F_\mu^{-1}(t_j) + F_\mu^{-1}(t_{j-1})}{2} \right)^2 \mathrm{d}t \tag{12}$$

$$\leq \frac{1}{4} \int_{I_j} \left[ (F_\mu^{-1}(t) - F_\mu^{-1}(t_j)) + (F_\mu^{-1}(t) - F_\mu^{-1}(t_{j-1})) \right]^2 \mathrm{d}t$$

$$\leq \frac{1}{4} \int_{I_j} \left[ (F_\mu^{-1}(t) - F_\mu^{-1}(t_{j-1})) - (F_\mu^{-1}(t_j) - F_\mu^{-1}(x)) \right]^2 \mathrm{d}t \tag{13}$$

$$\leq \frac{1}{4} \int_{I_j} \left( F_\mu^{-1}(t_j) - F_\mu^{-1}(t_{j-1}) \right)^2 \mathrm{d}t$$

$$\leq \frac{1}{4} |I_j| \left( F_\mu^{-1}(t_j) - F_\mu^{-1}(t_{j-1}) \right)^2,$$

where (12) follows from the fact that $\widehat{x}_j$ is the minimizer and (13) derives from observing that $F_\mu^{-1}(t) - F_\mu^{-1}(t_{j-1}) \leq 0$ by definition of $F_\mu^{-1}(t_{j-1})$ whereas $F_\mu^{-1}(t) - F_\mu^{-1}(t_{j-1}) \geq 0$. Let us rename $\Delta_j = F_\mu^{-1}(t_j) - F_\mu^{-1}(t_{j-1})$, the error can be bounded overall as:

$$W_2(\mu, \nu)^2 \leq \frac{1}{4} \sum_{j=1}^{M} |I_j| \Delta_j^2$$

$$\leq \frac{1}{4} \max_{j=1,2,\ldots,M} |I_j| \sum_{j=1}^{n} \Delta_j^2$$

$$\leq \frac{1}{4} \max_{j=1,2,\ldots,M} |I_j| \left( \sum_{j=1}^{M} \Delta_j \right)^2$$

$$\leq \frac{1}{4} \max_{j=1,2,\ldots,M} |I_j| \, (b-a)^2,$$

where we used Cauchy–Schwarz inequality in the last passage, observing that $\Delta_j \geq 0$ from the monotonicity of $F_\mu^{-1}$. $\qquad\square$

When we consider a uniform particle model, i.e., $w_j = 1/M$, the result reduces to:

$$W_2(\mu,\nu)^2 \leq \frac{(b-a)^2}{4M}. \tag{14}$$

The result tells us that when $M \to \infty$ the error vanishes as expected. Up to now, we considered the error introduced by representing a given distribution with a mixture of deltas. It is interesting to investigate the properties of the approximating distribution $\nu$.

**Lemma A.1.** *Let $\nu$ be the best mixture of deltas $L^2$-Wasserstein approximation of an arbitrary distribution $\mu$, as defined in Proposition A.2. If $\mu$ admits expectation, then it holds that:*

$$\mathbb{E}_{X\sim\mu}[X] = \mathbb{E}_{X\sim\nu}[X]. \tag{15}$$

*Proof.* We are going to assume that $\mu$ admits a p.d.f. $f_\mu$ and we indicate with $f_\nu$ the p.d.f. of $\nu$. We first observe that by making the substitution $x = F_\mu^{-1}(t)$, we have the identity:

$$\int_{I_j} F_\mu^{-1}(t)\mathrm{d}t = \int_{F_\mu^{-1}(t_{j-1})}^{F_\mu^{-1}(t_j)} x f_\mu(x)\mathrm{d}x. \tag{16}$$

Therefore:

$$\begin{aligned}
\mathbb{E}_{X\sim\nu}[X] &= \int_a^b x f_\nu(x)\mathrm{d}x \\
&= \sum_{j=1}^{M} w_j \widehat{x}_j \\
&= \sum_{j=1}^{M} w_j \frac{1}{|I_j|} \int_{I_j} F_\mu^{-1}(t)\mathrm{d}t \\
&= \sum_{j=1}^{M} \int_{I_j} F_\mu^{-1}(t)\mathrm{d}t \\
&= \sum_{j=1}^{M} \int_{F_\mu^{-1}(t_{j-1})}^{F_\mu^{-1}(t_j)} x f(x)\mathrm{d}x \\
&= \int_a^b x f_\mu(x)\mathrm{d}x = \mathbb{E}_{X\sim\mu}[X].
\end{aligned}$$

$\qquad\square$

Thus, our approximation preserves the mean, but unfortunately, this does not hold for the higher order moments.

## A.2  Combining Posteriors via Wasserstein Barycenters

In this section, we introduce the notion of Fréchet mean, we show how it reduces to the Wasserstein barycenter when selecting the Wasserstein distance as a metric.

**Definition A.1.** *Let $(\mathcal{X}, d)$ be a complete separable metric (Polish) space and $\mu \in \mathscr{P}_2(\mathcal{X})$ a probability measure over $\mathcal{X}$. The Fréchet mean is defined as:*

$$\overline{x} = \arg\inf_{x\in\mathcal{X}} \mathbb{E}_{Y\sim\mu}\left[d(x,Y)^2\right]. \tag{17}$$

The Fréchet mean generalizes the notion of mean by minimizing the expected value of a metric. In the particular case in which $\mathcal{X} = \mathbb{R}$, and $d(x,y) = |x - y|$ is the Euclidean distance, the Fréchet mean is the expectation of $X$ under $\mu$, i.e., $\overline{x} = \mathbb{E}_{X \sim \mu}[X]$.[9]

In our scenario, we have that $\mathcal{X} = \mathcal{Q}$ is a set of probability measures (the Q-posteriors) and we select as metric $d = W_2$, i.e., the $L^2$-Wasserstein distance. Therefore, Equation (17) defines the *Wasserstein barycenter* [2]. Typically, the notion of barycenter is defined in terms of a finite set of distributions [2]. However, we can also consider infinite (possibly continuous) sets of distributions. For our posterior distribution models, the Wasserstein barycenter is unique and we have a closed form expression.

### A.3    Closed-Forms for Wasserstein Barycenters of Gaussians and Particle models

We show in the following that the standard close forms for finite sets, when using Gaussian and particle models, naturally extends for continuous spaces.

**Proposition A.3.** *Let $(\mathcal{T}, \mathscr{F})$ be a measurable space and let $\mu \in \mathscr{P}(\mathcal{T})$ be a probability measure over $\mathcal{T}$. Let $\{\nu(t)\}_{t \in \mathcal{T}}$ be a family of probability measures. Then,*

$$\inf_{\nu \in \mathcal{N}} \mathbb{E}_{T \sim \mu} \left[ W_2\left(\nu, \nu(T)\right)^2 \right]$$

*has a unique solution both for Gaussian and uniform particle models. In particular, for the Gaussian model, the parameters of the $L^2$-Wasserstein barycenter are:*

$$\overline{m} = \mathbb{E}_{T \sim \mu}[\mu(T)], \quad \overline{\sigma} = \mathbb{E}_{T \sim \mu}[\sigma(T)],$$

*and for equally weighted particle models:*

$$\overline{x}_j = \mathbb{E}_{T \sim \mu}[x_j(T)], \quad j = 1, 2, ..., M.$$

*Proof.* All it takes is to write down the objective function and compute its minimizer. Let us start with the Gaussian model. The $L^2$-Wasserstein distance between two Gaussians is given in Equation (8), therefore our objective becomes:

$$\mathcal{L}(\mu, \sigma) = \mathbb{E}_{T \sim \mu} \left[ W_2\left(\nu, \nu(T)\right)^2 \right] = \mathbb{E}_{T \sim \mu} \left[ (\overline{m} - m(T))^2 + (\overline{\sigma} - \sigma(T))^2 \right],$$

where $\nu(T) = \mathcal{N}(m(T), \sigma^2(T))$ and $\overline{\nu} = \mathcal{N}(\overline{m}, \overline{\sigma}^2)$. The objective is clearly convex, therefore we just need to take the derivatives w.r.t. $\mu$ and $\sigma$:

$$\frac{\partial \mathcal{L}}{\partial \overline{\mu}} = 2 \mathbb{E}_{T \sim \mu}[\overline{m} - m(T)] = 0 \implies \overline{m} = \mathbb{E}_{T \sim \mu}[m(T)],$$

$$\frac{\partial \mathcal{L}}{\partial \overline{\sigma}} = 2 \mathbb{E}_{T \sim \mu}[\overline{\sigma} - \sigma(T)] = 0 \implies \overline{\sigma} = \mathbb{E}_{T \sim \mu}[\sigma(T)],$$

from which the result follows. Similarly, for the equally weighted particle models, using Proposition A.1, we have:

$$\mathcal{L}(x_1, x_2, ..., x_M) = \mathbb{E}_{T \sim \mu} \left[ W_2\left(\nu, \nu(T)\right)^2 \right]$$

$$= \mathbb{E}_{T \sim \mu} \left[ \sum_{j=1}^{M} w_j (\overline{x}_j - x_j(T))^2 \right]$$

$$= \sum_{j=1}^{M} w_j \mathbb{E}_{T \sim \mu} \left[ (\overline{x}_j - x_j(T))^2 \right].$$

We take the derivatives again and we obtain for every $j = 1, 2, ..., M$:

$$\frac{\partial \mathcal{L}}{\partial \overline{x}_j} = 2 w_j \mathbb{E}_{T \sim \mu}[\overline{x}_j - x_j(T)] = 0 \implies \overline{x}_j = \mathbb{E}_{T \sim \mu}[x_j(T)].$$

$\square$

In other words, the $L^2$-Wasserstein barycenter of a set of Gaussians is a Gaussian distribution having as mean the expectations of the mean and standard deviation the expectations of the standard deviations. Similarly, the $L^2$-Wasserstein barycenter of a set of uniform mixtures of deltas is a uniform mixture of deltas where each particle is located at the expectation of the locations.

## B Proofs and Derivations

**Proposition 3.1.** *If $\mathscr{D}$ is the set of deterministic distributions over $\mathbb{R}$, then the WTD update rule (Equation (5)) has a unique solution that corresponds to the TD update rule (Equation (1)).*

*Proof.* It is a simple application of Proposition A.3 for the particle model setting $M = 1$, $\mu = (1 - \alpha_t, \alpha_t)^T$, $\nu_1 = q_t(S_t, A_t)$ and $\nu_2 = R_{t+1} + \gamma v_t(S_{t+1})$. □

### B.1 Provable Efficiency

In this section, we provide a series of results about the provable efficiency of a slightly modified version of WQL. We restrict our attention to the Gaussian model and we consider the optimistic estimator (OE) for the maximum and optimistic exploration (OX). We will consider w.l.o.g. the following assumption.

**Assumption B.1.** *The reward function is deterministic and positive.*

As a consequence we have that $0 \le q_\pi(s, a) \le q_{\max}$. Several parts of the proofs we are going to present are inspired to [23]. We now illustrate how to modify WQL in order to have the desired theoretical guarantees.

#### B.1.1 Modified Gaussian WQL

First of all, we need to particularize the MWTD for the Gaussian case:

$$\widetilde{m}_{t+1}(s, a) = (1 - \alpha_t)\widetilde{m}_t(s, a) + \alpha_t \left(R_{t+1} + \gamma m_t(S_{t+1})\right),$$
$$\widetilde{\sigma}_{t+1}(s, a) = (1 - \alpha_t)\widetilde{\sigma}_t(s, a) + \alpha_t \gamma \sigma_t(S_{t+1}),$$
$$m_{t+1}(s, a) = \widetilde{m}_{t+1}(s, a) + \beta_t m_b,$$
$$\sigma_{t+1}(s, a) = \widetilde{\sigma}_{t+1}(s, a) + \beta_t \sigma_b,$$

with $m_b = 0$ by definition. We also define the auxiliary quantities that account for the accumulated effect of the learning rate:

$$\alpha_t^0 = \prod_{i=0}^t (1 - \alpha_i), \qquad \alpha_t^i = \alpha_i \prod_{j=i+1}^t (1 - \alpha_j)$$

It is clear that, by definition, for any $t = 0, 1, ...$ we have that $\alpha_0^t + \sum_{i=1}^t \alpha_i^t = 1$. Using these quantities we can rewrite the update rules for the mean and the standard deviation:

$$m_t(s, a) = \alpha_{n_t(s,a)} m_0 + \sum_{i=1}^{n_t(s,a)} \alpha_{n_t(s,a)}^i \left[R_{t_i+1} + \gamma m_{t_i}(S_{t_i+1})\right],$$

$$\sigma_t(s, a) = \alpha_{n_t(s,a)} \sigma_0 + \gamma \sum_{i=1}^{n_t(s,a)} \alpha_{n_t(s,a)}^i \sigma_{t_i}(S_{t_i+1}) + \beta_t \sigma_b,$$

where $n_t(s, a)$ is the number of times pair $(s, a)$ was visited up to time $t$, $t_i$ is the time at which pair $(s, a)$ was visited for the $i$-th time, $m_{t_i}(S_{t_i+1}) = m_{t_i}(S_{t_i+1}, \overline{a})$ and $\sigma_{t_i}(S_{t_i+1}, \overline{a})$, where $\overline{a} = \arg\max_{a \in \mathcal{A}} u_{t_i}^\delta(S_{t_i+1}, a)$ is the action that maximizes the upper bound of the Q-function, defined as:

$$\overline{u}_t^\delta(s, a) = m_t(s, a) + z_{1-\delta}\sigma_t(s, a), \quad u_t^\delta(s, a) = \min\left\{\overline{u}_t^\delta(s, a), q_{\max}\right\} \tag{18}$$

Notice that we can define an update rule for the upper bound $\overline{u}_t^\delta(s, a)$ too:

$$\overline{u}_t^\delta(s, a) = \alpha_{n_t(s,a)} \left[m_0 + z_{1-\delta}\sigma_0\right]$$

```
Input: m_0, σ_0, σ_b
1: for t = 1, 2, ... do
2:     Take action A_t ∈ arg max_{a∈A} u_t^δ(S_t, A_t)
3:     Observe S_{t+1} and R_{t+1}
4:     Update the posterior distribution
```
$$m_{t+1}(S_t, A_t) = (1 - \alpha_t)\widetilde{m}_t(S_t, A_t) + \alpha_t \left( R_{t+1} + \gamma m_t(S_{t+1}) \right)$$
$$\widetilde{\sigma}_{t+1}(S_t, A_t) = (1 - \alpha_t)\widetilde{\sigma}_t(S_t, A_t) + \alpha_t \gamma \sigma_t(S_{t+1})$$
$$\sigma_{t+1}(S_t, A_t) = \widetilde{\sigma}_{t+1}(S_t, A_t) + \beta_t \sigma_b$$
$$n_{t+1}(S_t, A_t) = n_t(S_t, A_t) + 1$$
```
5: end for
```

Algorithm 3: Modified Gaussian Wasserstein Q-Learning

$$+ \sum_{i=1}^{n_t(s,a)} \alpha_{n_t(s,a)}^i \left[ R_{t_i+1} + \gamma \left( m_{t_i}(S_{t_i+1}) + z_{1-\delta}\sigma_{t_i}(S_{t_i+1}) \right) \right] + \beta_t \sigma_b$$

$$= \alpha_{n_t(s,a)} \left[ m_0 + z_{1-\delta}\sigma_0 \right] + \sum_{i=1}^{n_t(s,a)} \alpha_{n_t(s,a)}^i \left[ R_{t_i+1} + \gamma u_t^\delta(S_{t_i+1}) \right] + \beta_t \sigma_b$$

where $u_t(S_{t_i+1})$ is the upper bound of the V-function defined as:

$$u_t^\delta(s) = \max_{a \in A} \left\{ u_t^\delta(s, a) \right\}. \tag{19}$$

In Algorithm 3 we provide the pseudocode of *Modified Wasserstein Q-Learning* (MWQL). The learning rates and the initialization values will be specified later in the analysis.

### B.1.2 Learning Rate

In this section, we introduce the learning rates $\alpha_t$ and $\beta_t$ we will use to prove our theoretical results and we will present some properties we are going to exploit for the subsequent proofs:

$$\alpha_t = \frac{a}{b+t}, \qquad \beta_t = \frac{c}{\sqrt{d+t}}. \tag{20}$$

with $0 < a \leq b + 1$, $b \geq 1$, $0 < c^2 \leq d$, $d \geq 1$ and $d \leq b$, whose values will be specified later. This choice of learning rates allows us to prove the following properties.

**Lemma B.1.** *If $a > 1$ and $b > 1$, the following relations hold for any $t = 1, 2, ...$ and $i = 1, 2, ...$:*

*1.* $\sum_{t=1}^{+\infty} \alpha_t^i \leq \frac{a}{a-1}$;

*2.* $\sum_{t=1}^{+\infty} \alpha_t^0 = \frac{b}{a-1} - 1$;

*3.* $\sum_{i=1}^{t} (\alpha_t^i)^2 \leq \frac{a}{b+t}$.

*Proof.* Let us start with 1. By using the properties of the Gamma function, we have that:

$$\sum_{t=1}^{M} \alpha_t^i = \frac{a}{a-1} \frac{\Gamma(b+i)}{\Gamma(1-a+b+i)} \left[ \frac{\Gamma(2-a+b)}{\Gamma(1+b)} - \frac{\Gamma(2-a+b+M)}{\Gamma(1+b+M)} \right].$$

Since $a > 1$ we have that $2 - a + b + M < 1 + b + M$, thus for $M \to +\infty$ the second addendum goes to zero. Moreover, for the same reason $\frac{\Gamma(b+i)}{\Gamma(1-a+b+i)} \leq 1$ and $\frac{\Gamma(2-a+b)}{\Gamma(1+b)} \leq 1$. The result follows immediately. A similar argument can be stated for 2. By using the properties of the Gamma function, we have the following equality:

$$\sum_{t=1}^{M} \alpha_t^0 = \frac{b}{a-1} - 1 - \frac{\Gamma(1+b)\Gamma(2-a+b+M)}{(a-1)\Gamma(1-a+b)\Gamma(1+b+M)}.$$

Since $a > 1$ we have that $2 - a + b + M < 1 + b + M$ and therefore for $M \to +\infty$ the ratio of Gamma functions goes to zero, from which the result follows. Finally, for 3 we employ an argument similar to that of Lemma 4.1 (b) of [23]:

$$
\begin{aligned}
\alpha_t^i &= \frac{a}{b+i} \left( \frac{b+i+1-a}{b+i+1} \cdot \frac{b+i+2-a}{b+i+2} \cdots \frac{b+t-a}{b+t} \right) \\
&= \frac{a}{b+t} \left( \frac{b+i+1-a}{b+i} \cdot \frac{b+i+2-a}{b+i+1} \cdots \frac{b+t-a}{b+t-1} \right) \\
&\leq \frac{a}{b+t},
\end{aligned}
$$

where we exploited the fact that $\frac{b+i+j-a}{b+i+j-1} \leq 1$ for all $j = 1, ... t - i$ being $a > 1$. Now, observing that $\sum_{i=1}^t \alpha_t^i \leq 1$, we have:

$$
\sum_{i=1}^t (\alpha_t^i)^2 \leq \frac{a}{b+t} \sum_{i=1}^t \alpha_t^i \leq \frac{a}{b+t}.
$$

$\square$

### B.1.3 Optimism

We now prove that with a suitable choice of $m_0$, $\sigma_0$ and $\sigma_b$ we are able to guarantee that $u_t^\delta(s,a)$ is optimistic w.r.t. $q^*(s,a)$ with high probability. We start proving the following intermediate result.

**Lemma B.2.** *For any $\delta \in [0,1]$, with probability at least $1 - \delta$, we have simultaneously for all $s \in \mathcal{S}$, $a \in \mathcal{A}$ and $t \in \{1, 2, ..., T\}$:*

$$
\sum_{i=1}^{n_t(s,a)} \alpha_{n_t(s,a)}^i \left[ u_{t_i}^\delta(S_{t_i+1}) - \mathop{\mathbb{E}}_{S' \sim \mathcal{P}(\cdot|s,a)} \left[ u_{t_i}^\delta(S') \right] \right] \leq q_{\max} \sqrt{\sum_{i=1}^{n_t(s,a)} \left( \alpha_{n_t(s,a)}^i \right)^2 \log \frac{|\mathcal{S}||\mathcal{A}|T}{\delta}}.
$$

*Proof.* Let us provide a formal definition of $t_i$:

$$
t_i = \min \left( \{ t \in \{1, 2, ..., T\} : t > t_{i-1} \land (s_t, a_t) = (s, a) \} \cup \{T + 1\} \right), \tag{21}
$$

where we have assigned the value $T + 1$ if $(s, a)$ is experienced less than $i$ times. Consider the filtration $\mathcal{F}_i = \sigma(S_0, A_0, R_1, ..., S_{t_i-1}, A_{t_i-1}, R_{t_i})$ generated by all the random variables realized until time $t_i$. The random variable $X_{t_i} = \mathbb{1}_{\{t_i \leq T\}} \left[ u_{t_i}^\delta(S_{t_i+1}) - \mathbb{E}_{S' \sim \mathcal{P}(\cdot|s,a)} \left[ u_{t_i}^\delta(S') \right] \right]$ is a martingale difference sequence (MDS) w.r.t. the filtration $\{\mathcal{F}_i\}_{i=1,2,...}$, as $\mathbb{E}[X_{t_i}|\mathcal{F}_i] = 0$ and $|X_{t_i}| < q_{\max}$ a.s.. Using Azuma-Hoeffding inequality and a union bound over the time $\{1, 2, ..., T\}$, the states $\mathcal{S}$ and the actions $\mathcal{A}$ we have that w.p. at least $1 - \delta$ the statement holds. $\square$

We are now ready to prove that $u_t^\delta(s,a)$ are optimistic with high probability.

**Theorem B.1.** *(Optimism) Let $m_0 = q_{\max}$, $\sigma_0 = 0$ and $\sigma_b = \frac{\gamma q_{\max}}{cz_{1-\delta}} \sqrt{a \log \frac{|\mathcal{S}||\mathcal{A}|T}{\delta}}$ for all $s \in \mathcal{S}$, $a \in \mathcal{A}$. Then, for any $\delta \in [0,1]$, with probability at least $1 - \delta$, we have simultaneously for all $s \in \mathcal{S}$, $a \in \mathcal{A}$ and $t \in \{1, 2, ..., T\}$:*

$$
u_t^\delta(s,a) \geq q^*(s,a). \tag{22}
$$

*Proof.* The proof is by induction on $t$. For $t = 0$, we have that $u_0^\delta(s,a) \geq q_{\max} \geq q^*(s,a)$. Let us assume the statement hold up to time $t - 1$, we prove that it holds for $t$. Recall that $u_t^\delta(s,a) = \min\{\overline{u}_t^\delta(s,a), q_{\max}\}$. If $u_t^\delta(s,a) = q_{\max}$ then the statement hold. Otherwise we have $u_t^\delta(s,a) = \overline{u}_t^\delta(s,a)$. Let us write explicitly the expression of the upper bound $\overline{u}_t^\delta(s,a)$. With probability at least $1 - \delta$ we have:

$$
\begin{aligned}
\overline{u}_t^\delta(s,a) &= \alpha_{n_t(s,a)} \left[ m_0 + z_{1-\delta}\sigma_0 \right] + \sum_{i=1}^{n_t(s,a)} \alpha_{n_t(s,a)}^i \left[ R_{t_i+1} + \gamma u_t^\delta(S_{t_i+1}) \right] + \beta_{n_t(s,a)} z_{1-\delta}\sigma_b \\
&= \alpha_{n_t(s,a)} \left[ m_0 + z_{1-\delta}\sigma_0 \right] + \beta_{n_t(s,a)} z_{1-\delta}\sigma_b \\
&\quad + \sum_{i=1}^{n_t(s,a)} \alpha_{n_t(s,a)}^i \left[ r(s,a) + \gamma \mathop{\mathbb{E}}_{S' \sim \mathcal{P}(\cdot|s,a)} \left[ u_{t_i}^\delta(S') \right] \right] \\
&\quad + \gamma \sum_{i=1}^{n_t(s,a)} \alpha_{n_t(s,a)}^i \left[ u_{t_i}^\delta(S_{t_i+1}) - \mathop{\mathbb{E}}_{S' \sim \mathcal{P}(\cdot|s,a)} \left[ u_{t_i}^\delta(S') \right] \right]
\end{aligned} \tag{23}
$$

$$\geq \alpha^0_{n_t(s,a)} q^*(s,a) + \sum_{i=1}^{n_t(s,a)} \alpha^i_{n_t(s,a)} \left[ r(s,a) + \gamma \mathop{\mathbb{E}}_{S' \sim \mathcal{P}(\cdot|s,a)} \left[ v^*(S') \right] \right] \tag{24}$$

$$+ \beta_{n_t(s,a)} z_{1-\delta} \sigma_b - \gamma q_{\max} \sqrt{\sum_{i=1}^{n_t(s,a)} \left( \alpha^i_{n_t(s,a)} \right)^2 \log \frac{|\mathcal{S}||\mathcal{A}|T}{\delta}} \tag{25}$$

$$\geq q^*(s,a) + \beta_{n_t(s,a)} z_{1-\delta} \sigma_b - \gamma q_{\max} \sqrt{\frac{a}{b + n_t(s,a)} \log \frac{|\mathcal{S}||\mathcal{A}|T}{\delta}}, \tag{26}$$

where line (23) derives from the fact that the reward is deterministic ($R_{t_i+1} = r(s,a)$), line (24) is an application of the inductive hypothesis being all $t_i < t$ and observing that $u^\delta_{t_i}(S') = \max_{a \in \mathcal{A}} \{ u^\delta_{t_i}(S', a) \} \geq \max_{a \in \mathcal{A}} \{ q^*(S', a) \} = v^*(S')$ and we applied Lemma B.2 at line (25). Line (26) follows from the application of Bellman equation and using Lemma B.1 to bound the summation in the square root. In order to guarantee that this expression is non-negative for all $t \in \{1, 2, ..., T\}$ we need to satisfy:

$$\beta_{n_t(s,a)} z_{1-\delta} \sigma_b \geq \gamma q_{\max} \sqrt{\frac{a}{b + n_t(s,a)} \log \frac{|\mathcal{S}||\mathcal{A}|T}{\delta}}$$

$$\implies \sigma_b \geq \gamma q_{\max} \frac{\sqrt{d + n_t(s,a)}}{c z_{1-\delta}} \sqrt{\frac{a}{b + n_t(s,a)} \log \frac{|\mathcal{S}||\mathcal{A}|T}{\delta}}.$$

The term depending on $t$ can be bounded recalling that $d \leq b$ and that $n_t(s,a) \leq T$:

$$\sqrt{\frac{d + n_t(s,a)}{b + n_t(s,a)}} \leq \sqrt{\frac{d + T}{b + T}} \leq 1.$$

Therefore, we choose:

$$\sigma_b = \frac{\gamma q_{\max}}{c z_{1-\delta}} \sqrt{a \log \frac{|\mathcal{S}||\mathcal{A}|T}{\delta}}.$$

$\square$

### B.1.4 Main Result

We now provide this central lemma that we will use to state a bound on the sample complexity considering our running algorithm as a non stationary policy $\mathfrak{A}$.

**Lemma B.3.** *Let $s \in \mathcal{S}$ be any state and let $\Delta_t(s) = v^*(s) - v_{\mathfrak{A}}(s)$ be the instantaneous regret of state $s$ at time $t$ and define $\Psi_t(s) = u^\delta_t(s) - v_{\mathfrak{A}}(s)$. Let $\delta \in [0, 1]$, then with probability at least $1 - \delta$ the following chain inequalities holds simultaneously for all $s \in \mathcal{S}$, $a \in \mathcal{A}$ and $t \in \{1, 2, ..., T\}$: $\Delta_t(s) \leq \Psi_t(s)$ and*

$$\Psi_t(s) \leq \alpha^0_{n_t(s,a)} q_{\max} + \gamma \sum_{i=1}^{n_t(s,a)} \alpha^i_{n_t(s,a)} \Psi_{t_i}(s_{t_i+1}) + 2\gamma q_{\max} \sqrt{\frac{a}{d + n_t(s,a)} \log \frac{|\mathcal{S}||\mathcal{A}|T}{\delta}}, \tag{27}$$

*where $a \in \arg\max_{a' \in \mathcal{A}} \{ u^\delta_t(s, a') \}$.*

*Proof.* Consider a state $s \in \mathcal{S}$, we decompose the instantaneous regret at time $t$, i.e., $\Delta_t(s)$. It is important to notice that $s$ is not necessarily the state visited by our algorithm at time $t$, i.e., $s_t$. With probability at least $1 - \delta$ we have simultaneously for all $t$:

$$\Delta_t(s) = v^*(s) - v_{\mathfrak{A}}(s)$$

$$= \max_{a \in \mathcal{A}} \{ q^*(s, a) \} - v_{\mathfrak{A}}(s)$$

$$\leq \max_{a' \in \mathcal{A}} \{ u^\delta_t(s, a') \} - v_{\mathfrak{A}}(s) \tag{28}$$

$$= u^\delta_t(s) - v_{\mathfrak{A}}(s) = \Psi_t(s) \tag{29}$$

$$\leq u^\delta_t(s, a) - q_{\mathfrak{A}}(s, a), \tag{30}$$

were $a \in \arg\max_{a' \in \mathcal{A}} u^\delta_t(s, a')$. Line (28) follows from the optimism (Theorem B.1) and line (30) is obtained by observing that $\max_{a' \in \mathcal{A}} \{ q_{\mathfrak{A}}(s, a') \} \geq q_{\mathfrak{A}}(s, a)$ for any $a \in \mathcal{A}$. We now apply the Bellman equation on the upper confidence bound:

$$u^\delta_t(s, a) - q_{\mathfrak{A}}(s, a) \leq \overline{u}^\delta_t(s, a) - q_{\mathfrak{A}}(s, a) \tag{31}$$

$$= \alpha_{n_t(s,a)}^0 \left( m_0 - q_{\mathfrak{A}}(s,a) \right) + \beta_{n_t(s,a)} z_{1-\delta} \sigma_b$$

$$+ \sum_{i=1}^{n_t(s,a)} \gamma \alpha_{n_t(s,a)}^i \left( u_{t_i}^\delta (s_{t_i+1}) - \mathop{\mathbb{E}}_{S' \sim \mathcal{P}(\cdot | s,a)} \left[ v_{\mathfrak{A}}(S') \right] \right)$$

$$= \alpha_{n_t(s,a)}^0 \left( m_0 - q_{\mathfrak{A}}(s,a) \right) + \beta_{n_t(s,a)} z_{1-\delta} \sigma_b$$

$$+ \gamma \sum_{i=1}^{n_t(s,a)} \alpha_{n_t(s,a)}^i \left( u_{t_i}^\delta (s_{t_i+1}) - v_{\mathfrak{A}}(s_{t_i+1}) \right)$$

$$+ \gamma \sum_{i=1}^{n_t(s,a)} \alpha_{n_t(s,a)}^i \left( v_{\mathfrak{A}}(s_{t_i+1}) - \mathop{\mathbb{E}}_{S' \sim \mathcal{P}(\cdot | s,a)} \left[ v_{\pi_t}(S') \right] \right)$$

$$\leq \alpha_{n_t(s,a)}^0 q_{\max} + \gamma q_{\max} \sqrt{\frac{a}{d + n_t(s,a)} \log \frac{|\mathcal{S}||\mathcal{A}|T}{\delta}} \tag{32}$$

$$+ \gamma \sum_{i=1}^{n_t(s,a)} \alpha_{n_t(s,a)}^i \left( u_{t_i}^\delta (S_{t_i+1}) - v_{\mathfrak{A}}(S_{t_i+1}) \right)$$

$$+ \gamma q_{\max} \sqrt{\sum_{i=1}^{n_t(s,a)} \left( \alpha_{n_t(s,a)}^i \right)^2 \log \frac{|\mathcal{S}||\mathcal{A}|T}{\delta}} \tag{33}$$

$$\leq \alpha_{n_t(s,a)}^0 q_{\max} + 2\gamma q_{\max} \sqrt{\frac{a}{d + n_t(s,a)} \log \frac{|\mathcal{S}||\mathcal{A}|T}{\delta}} + \gamma \sum_{i=1}^{n_t(s,a)} \alpha_{n_t(s,a)}^i \Psi_{t_i}(S_{t_i+1}),$$

where line (32) follows from observing that $m_0 - q_{\mathfrak{A}}(s,a) = q_{\max} - q_{\mathfrak{A}}(s,a) \leq q_{\max}$ and by substitution of the value of $\sigma_0^{(2)}(s,a)$ and line (33) is obtained by applying Azuma-Hoeffding inequality, like in Lemma B.2 (all it takes it to consider $v_{\mathfrak{A}}$ instead of $u_{t_i}^\delta$) and recalling that $d \leq b$. $\qquad\square$

Using the previous lemma we are able to state the following theorem that represents the core of our analysis. In this case, we are going to evaluate how well is our algorithm performing (in terms of value function) over the states visited by the algorithm itself. This will allow us to derive immediately a guarantee on the sample complexity of PE-WQL.

**Theorem B.2.** *Let $S_0, S_1, ..., S_T$ be the sequence of states and actions visited by the algorithm. Let $a = \frac{2+\gamma}{2(1-\gamma)}$ and $b = a - 1$. Then, under the same assumptions as Lemma B.1, for any $\delta \in [0,1]$, with probability at least $1 - \delta$ it holds that:*

$$\sum_{t=1}^T \Delta_t(S_t) \leq \mathcal{O} \left( \frac{q_{\max}}{(1-\gamma)^{3/2}} \sqrt{|\mathcal{S}||\mathcal{A}|T \log \frac{|\mathcal{S}||\mathcal{A}|T}{\delta}} \right). \tag{34}$$

*Proof.* We are now going do deal with the summation $\sum_{t=1}^T \Psi_t(s_{t+1})$:

$$\sum_{t=1}^T \Psi_t(s_{t+1}) = q_{\max} \underbrace{\sum_{t=1}^T \alpha_{n_t(s_{t+1},a_{t+1})}^0}_{\text{(i)}} + \gamma q_{\max} \sqrt{a \log \frac{|\mathcal{S}||\mathcal{A}|T}{\delta}} \underbrace{\sum_{t=1}^T \frac{1}{\sqrt{d + n_t(s_{t+1},a_{t+1})}}}_{\text{(ii)}}$$

$$+ \gamma \underbrace{\sum_{t=1}^T \sum_{i=1}^{n_t(s_{t+1},a_{t+1})} \alpha_{n_t(s_{t+1},a_{t+1})}^i \Psi_{t_i(s_{t+1},a_{t+1})} \left( s_{t_i(s_{t+1},a_{t+1})+1} \right)}_{\text{(iii)}}$$

We make the following observation we are going to use throughout the proof

$$n_t(s_{t+1}, a_{t+1}) = n_{t+1}(s_{t+1}, a_{t+1}) - 1. \tag{35}$$

Let us start with (i):

$$\sum_{t=1}^T \alpha_{n_t(s_{t+1},a_{t+1})}^0 = \sum_{t=1}^T \alpha_{n_{t+1}(s_{t+1},a_{t+1})-1}^0$$

$$= \sum_{h=2}^{T+1} \alpha_{n_h(s_h,a_h)-1}^0 \tag{36}$$

$$\leq \sum_{h=1}^{T+1} \alpha^0_{n_h(s_h,a_h)-1}$$

$$= \sum_{s\in\mathcal{S}} \sum_{a\in\mathcal{A}} \sum_{i=1}^{n_{T+1}(s,a)} \alpha^0_{i-1} \tag{37}$$

$$= \sum_{s\in\mathcal{S}} \sum_{a\in\mathcal{A}} \sum_{i=0}^{n_{T+1}(s,a)-1} \alpha^0_i$$

$$\leq \sum_{s\in\mathcal{S}} \sum_{a\in\mathcal{A}} \sum_{i=0}^{+\infty} \alpha^0_i = \left(\frac{b}{a-1}+1\right)|\mathcal{S}||\mathcal{A}|, \tag{38}$$

where we made the change of variable $h = t+1$ to get line (36), we decomposed the summation over the state action pairs obseving that each of them appears $n_{T+1}(s,a)$ times to get line (37) and we used Lemma B.1 to get line (38).

Let us now consider (ii); using Equation (35) we get:

$$\sum_{t=1}^{T} \frac{1}{\sqrt{d+n_t(s_{t+1},a_{t+1})}} = \sum_{t=1}^{T} \frac{1}{\sqrt{d+n_{t+1}(s_{t+1},a_{t+1})-1}}$$

$$= \sum_{h=2}^{T+1} \frac{1}{\sqrt{d+n_h(s_h,a_h)-1}}$$

$$\leq \sum_{h=1}^{T+1} \frac{1}{\sqrt{d+n_h(s_h,a_h)-1}}$$

$$= \sum_{s\in\mathcal{S}} \sum_{a\in\mathcal{A}} \sum_{i=1}^{n_{T+1}(s,a)} \frac{1}{\sqrt{d+i-1}} \tag{39}$$

$$\leq 2 \sum_{s\in\mathcal{S}} \sum_{a\in\mathcal{A}} \sqrt{n_{T+1}(s,a)} \tag{40}$$

$$\leq 2\sqrt{|\mathcal{S}||\mathcal{A}|(T+1)}, \tag{41}$$

where line (39) derives from decomposing the summation over state-action pairs and observing that each state-action pair appears $n_{T+1}(s,a)$ times. Line (40) is obtained by bounding the summation with the integral: $\int_1^{n_{T+1}(s,a)+1} \frac{1}{\sqrt{b+x-1}}dx = 2\sqrt{d+n_{T+1}(s,a)} - 2\sqrt{d} \leq 2\sqrt{n_{T+1}(s,a)}$, where the last inequality derives from the subadditivity of the square root. Finally, line (41) is obtained by observing that $\sum_{s\in\mathcal{S}} \sum_{a\in\mathcal{A}} n_{T+1}(s,a) = T+1$ and the expression is maximized by taking $n_{T+1}(s,a) = \frac{T+1}{|\mathcal{S}||\mathcal{A}|}$.

Now we consider the term (iii). First observe that $n_t(s_{t+1},a_{t+1}) \geq 1$ in order to appear in the inner summation. Consider the derivation:

$$\sum_{t=1}^{T} \sum_{i=1}^{n_t(s_{t+1},a_{t+1})} \alpha^i_{n_t(s_{t+1},a_{t+1})} \Psi_{t_i(s_{t+1},a_{t+1})}(s_{t_i(s_{t+1},a_{t+1})+1})$$

$$= \sum_{t=1}^{T} \sum_{i=1}^{n_{t+1}(s_{t+1},a_{t+1})-1} \alpha^i_{n_{t+1}(s_{t+1},a_{t+1})-1} \Psi_{t_i(s_{t+1},a_{t+1})}(s_{t_i(s_{t+1},a_{t+1})+1})$$

$$= \sum_{h=2}^{T+1} \sum_{i=1}^{n_h(s_h,a_h)-1} \alpha^i_{n_h(s_h,a_h)-1} \Psi_{t_i(s_h,a_h)}(s_{t_i(s_h,a_h)+1})$$

$$\leq \sum_{h=1}^{T+1} \sum_{i=1}^{n_h(s_h,a_h)-1} \alpha^i_{n_h(s_h,a_h)-1} \Psi_{t_i(s_h,a_h)}(s_{t_i(s_h,a_h)+1}) \tag{42}$$

$$= \sum_{t=1}^{T+1} \Psi_t(s_{t+1}) \sum_{i=n_t(s_t,a_t)}^{n_{T+1}(s_t,a_t)} \alpha^{n_t(s_t,a_t)}_{i-1} \tag{43}$$

$$\leq \sum_{t=1}^{T+1} \Psi_t(s_{t+1}) \sum_{i=2}^{+\infty} \alpha^{n_t(s_t,a_t)}_{i-1} \tag{44}$$

$$\leq \sum_{t=1}^{T+1} \Psi_t(s_{t+1}) \sum_{i=1}^{+\infty} \alpha_i^{n_t(s_t,a_t)}$$

$$\leq \frac{a}{a-1} \sum_{t=1}^{T+1} \Psi_t(s_{t+1}) \tag{45}$$

$$\leq \frac{a}{a-1} \sum_{t=1}^{T} \Psi_t(s_{t+1}) + \frac{a}{a-1} q_{\max}. \tag{46}$$

where line (43) is obtained by observing that given $\Psi_t(s_{t+1})$ it is going to appear in the summation for all $t \geq t_i(s_t, a_t)$. The first time it will appear multiplied by $\alpha_{n_t(s_t,a_t)-1}^{n_t(s_t,a_t)}$, the second time with $\alpha_{n_t(s_t,a_t)}^{n_t(s_t,a_t)}$ and so on. Line (44) derives from observing that $n_t(s_{t+1}, a_{t+1}) \geq 1$, thus $n_{t+1}(s_{t+1}, a_{t+1}) \geq 2$. Line (45) is obtained by applying Lemma B.1.

Now, we put all together into the bound on the summation:

$$\sum_{t=1}^{T} \Psi_t(s_{t+1}) \leq q_{\max}\left(\frac{b}{a-1}+1\right)|\mathcal{S}||\mathcal{A}| + 2\gamma q_{\max}\sqrt{a \log \frac{|\mathcal{S}||\mathcal{A}|T}{\delta}}\sqrt{|\mathcal{S}||\mathcal{A}|(T+1)}$$

$$+ \frac{a\gamma}{a-1}\sum_{t=1}^{T}\Psi_t(s_{t+1}) + \frac{a\gamma}{a-1}q_{\max}.$$

In order to solve the inequality, we must require $\frac{a\gamma}{a-1} < 1$, i.e., $a > \frac{1}{1-\gamma}$. In such a case, we obtain:

$$\sum_{t=1}^{T}\Psi_t(s_{t+1}) \leq \left(1 - \frac{a\gamma}{1-a}\right)^{-1}\left[q_{\max}\left(\frac{b}{a-1}+1\right)|\mathcal{S}||\mathcal{A}|\right.$$

$$\left. + 2\gamma q_{\max}\sqrt{a \log \frac{|\mathcal{S}||\mathcal{A}|T}{\delta}}\sqrt{|\mathcal{S}||\mathcal{A}|(T+1)} + \frac{a\gamma}{a-1}q_{\max}\right].$$

Now, we propose a value for $a$ and $b$ that fulfills all the conditions. In particular, among all possible values we select:[10] $a = \frac{2+\gamma}{2(1-\gamma)}$. Then we take the smallest possible value for $b$, i.e., $b = a - 1$. From which we get:

$$\sum_{t=1}^{T}\Psi_t(s_{t+1}) \leq \frac{3}{1-\gamma}\left[2q_{\max}|\mathcal{S}||\mathcal{A}| + 2\gamma q_{\max}\sqrt{\frac{2+\gamma}{2(1-\gamma)}\log\frac{|\mathcal{S}||\mathcal{A}|T}{\delta}}\sqrt{|\mathcal{S}||\mathcal{A}|(T+1)} + \frac{2+\gamma}{3}q_{\max}\right]$$

$$\leq \frac{3}{1-\gamma}\left[q_{\max}(|\mathcal{S}||\mathcal{A}|+1) + 2\gamma q_{\max}\sqrt{\frac{3}{2(1-\gamma)}\log\frac{|\mathcal{S}||\mathcal{A}|T}{\delta}}\sqrt{|\mathcal{S}||\mathcal{A}|(T+1)}\right]$$

$$\leq \mathcal{O}\left(\frac{q_{\max}}{(1-\gamma)^{3/2}}\sqrt{|\mathcal{S}||\mathcal{A}|T \log\frac{|\mathcal{S}||\mathcal{A}|T}{\delta}}\right),$$

where the last passage is obtained by observing that: if $T + 1 \geq \sqrt{|\mathcal{S}||\mathcal{A}|(T+1)}$ then $\sqrt{|\mathcal{S}||\mathcal{A}|(T+1)} \geq |\mathcal{S}||\mathcal{A}|$; on the contrary $\sum_{t=1}^{T}\Psi_t(s_{t+1}) \leq T \leq \sqrt{|\mathcal{S}||\mathcal{A}|(T+1)} - 1$. Thus, we can discard the first term. $\square$

Theorem B.2 allows us to bound the per-step regret over the trajectories visited by the algorithm.

**Corollary B.1.** *Let $a = \frac{2+\gamma}{2(1-\gamma)}$ and $b = a - 1$. Then, under the same assumptions as Lemma B.1, for any $\delta \in [0,1]$, with probability at least $1 - \delta$ for $T \geq T_0$:*

$$T_0 = \mathcal{O}\left(\frac{q_{\max}^2|\mathcal{S}||\mathcal{A}|}{\epsilon^2(1-\gamma)^3}\log\frac{q_{\max}^2|\mathcal{S}|^2|\mathcal{A}|^2}{\delta\epsilon^2(1-\gamma)^3}\right)$$

*we have that:*

$$\frac{1}{T}\sum_{t=1}^{T}\Delta_t(S_t) \leq \epsilon.$$

*Proof.* Following the reasoning of [21], we just need to find a sufficiently large $T_0$ such that for all $T \geq T_0$ the per step regret is smaller than $\epsilon$:

$$\frac{1}{T} \mathcal{O}\left( \frac{q_{\max}}{(1-\gamma)^{3/2}} \sqrt{|\mathcal{S}||\mathcal{A}|T \log \frac{|\mathcal{S}||\mathcal{A}|T}{\delta}} \right) < \epsilon \implies T \geq \mathcal{O}\left( \frac{q_{\max}^2 |\mathcal{S}||\mathcal{A}|}{\epsilon^2 (1-\gamma)^3} \log \frac{|\mathcal{S}||\mathcal{A}|T}{\delta} \right)$$

We rename $\tau = \frac{q_{\max}^2 |\mathcal{S}||\mathcal{A}|}{\epsilon^2 (1-\gamma)^3}$. We select $T_0 = \mathcal{O}\left( 2\tau \log \frac{\tau |\mathcal{S}||\mathcal{A}|}{\delta} \right)$. Therefore, we have:

$$\begin{aligned}
T_0 &= \mathcal{O}\left( 2\tau \log \frac{\tau |\mathcal{S}||\mathcal{A}|}{\delta} \right) \\
&= \mathcal{O}\left( \tau \log \left( \frac{\tau |\mathcal{S}||\mathcal{A}|}{\delta} \frac{\tau |\mathcal{S}||\mathcal{A}|}{\delta} \right) \right) \\
&\geq \mathcal{O}\left( \tau \log \left( 2 \frac{\tau |\mathcal{S}||\mathcal{A}|}{\delta} \log \frac{\tau |\mathcal{S}||\mathcal{A}|}{\delta} \right) \right) \\
&= \mathcal{O}\left( \tau \log \frac{T_0 |\mathcal{S}||\mathcal{A}|}{\delta} \right) \\
&= \mathcal{O}\left( \frac{q_{\max}^2 |\mathcal{S}||\mathcal{A}|}{\epsilon^2 (1-\gamma)^3} \log \frac{|\mathcal{S}||\mathcal{A}|T_0}{\delta} \right),
\end{aligned}$$

where we exploited the inequality $x > 2 \log x$ for $x > 0$. $\square$

Finally, we prove that MWQL is PAC-MDP in the average loss setting. We import several ideas from [41]. First of all, we recall the notion of adjusted loss.

**Definition B.1** (Definition 5 of [42]). *Suppose a learning algorithm $\mathfrak{A}$ is run for one sequence of $T_1 + T_2 - 1$ steps. Consider partial sequence $S_0, R_1, ..., S_{t-1}, R_t, S_t$ visited by $\mathfrak{A}$. For any policy $\pi$ and integer $t$ such that $t \leq T_1$, let $R_{T_2}^\pi(t) = \sum_{t'=t}^{t+T_2-1} \gamma^{t'-t} R_{t'+1} + \gamma^{T_2} v_\pi(S_{t+T_2})$ be the* adjusted return. *Let $I_{T_2}^\pi(t) = v_\pi(S_t) - R_{T_2}^\pi(t)$ be the* adjusted instantaneous loss. *Let $L_{T_1,T_2}^\pi = \frac{1}{T_1} \sum_{t=1}^{T_1} I_{T_2}^\pi(t)$ be the* adjusted average loss.

We are now ready to prove the result.

**Theorem 5.2.** *Under the hypothesis of Theorem 5.1, MWQL with Gaussian posterior, OE and OX is PAC-MDP in the average loss setting, i.e., for any $\epsilon \geq 0$ and $\delta \in [0,1]$, after*

$$T = \mathcal{O}\left( \frac{q_{\max}^2 |\mathcal{S}||\mathcal{A}|}{\epsilon^2 (1-\gamma)^3} \log \frac{q_{\max}^2 |\mathcal{S}|^2 |\mathcal{A}|^2}{\delta \epsilon^2 (1-\gamma)^3} \right)$$

*steps we have that the average loss $\mathcal{L}_\mathfrak{A} \leq \epsilon$ with probability at least $1 - \delta$.*

*Proof.* A running algorithm can be viewed as a non stationary policy $\mathfrak{A}$. Define $T = T_1 + T_2 - 1$ and define the adjusted average loss of $\mathfrak{A}_t$ w.r.t. itself.

$$L_{T_1,T_2}^\mathfrak{A} = \frac{1}{T_1} \sum_{t=1}^{T_1} I_{T_2}^\mathfrak{A}(t). \tag{47}$$

In [42] it is proven that for any algorithm $\mathfrak{A}$ and for

$$T_1 \geq \max \left\{ \frac{1 + 2\log(1/\delta) q_{\max}^2}{\epsilon^2 (1-\gamma)^2}, 2\log(1/\delta) q_{\max}^2 (T_2 - 1) \right\}, \tag{48}$$

we have that $L_{T_1,T_2}^\mathfrak{A} \leq \epsilon$ with probability at least $1 - \delta$. We now consider the adjusted average loss w.r.t. to the optimal policy $\pi^*$ and we decompose it:

$$L_{T_1,T_2}^{\pi^*} = \frac{1}{T_1} \sum_{t=1}^{T_1} I_{T_2}^{\pi^*}(t) \tag{49}$$

$$= \frac{1}{T_1} \sum_{t=1}^{T_1} I_{T_2}^{\pi^*}(t) \pm \frac{1}{T_1} \sum_{t=1}^{T_1} I_{T_2}^\mathfrak{A}(t)$$

$$\leq \epsilon + \frac{1}{T_1} \sum_{t=1}^{T_1} \left( I_{T_2}^{\pi^*}(t) - I_{T_2}^\mathfrak{A}(t) \right) \tag{50}$$

$$= \epsilon + \frac{1}{T_1} \sum_{t=1}^{T_1} \left( v_{\pi^*}(S_t) - R_{T_2}^{\pi^*}(t) - \left( v_{\mathfrak{A}}(S_t) - R_{T_2}^{\mathfrak{A}}(t) \right) \right) \tag{51}$$

$$= \epsilon + \frac{1}{T_1} \sum_{t=1}^{T_1} (v_{\pi^*}(S_t) - v_{\mathfrak{A}}(S_t)) + \frac{1}{T_1} \sum_{t=1}^{T_1} \gamma^{T_2} (v_{\mathfrak{A}}(S_{t+T_2}) - v_{\pi^*}(S_{t+T_2})) \tag{52}$$

$$\leq \epsilon + \frac{1}{T_1} \sum_{t=1}^{T_1} \Delta_t(S_t) \tag{53}$$

$$\leq 2\epsilon, \tag{54}$$

where (50) derives from the fact that $L_{T_1,T_2}^{\mathfrak{A}} \leq \epsilon$, (51) is from the definition of adjusted instantaneous loss, (52) derives from the definition of adjusted return, (53) is obtained by observing that $v_{\mathfrak{A}}(S_{t+T_2}) \leq v_{\pi^*}(S_{t+T_2})$ and the definition of $\Delta_t(S_t)$ and (54) derives from Corollary B.1. Therefore, the inequality hold for $T_1$ satisfying condition (48) and Corollary B.1, with probability at least $1 - 2\delta$. Proposition 3 of [42] proves that for $T_1 \geq \frac{2T_2}{\epsilon}$ and $T_2 \geq \frac{\log(\epsilon(1-\gamma))}{\log \gamma}$ we have that the adjusted loss $L_{T_1,T_2}^{\pi^*}$ is $\epsilon$-close to the average loss $\mathcal{L}_{\mathfrak{A}}$. Therefore we have that $\mathcal{L}_{\mathfrak{A}} \leq 3\epsilon$ with probability at least $1 - 2\delta$ provided that:

$$T_1 = \mathcal{O}\left( \frac{q_{\max}^2 |\mathcal{S}||\mathcal{A}|}{9\epsilon^2(1-\gamma)^3} \log \frac{q_{\max}^2 |\mathcal{S}|^2 |\mathcal{A}|^2}{2\delta\epsilon^2(1-\gamma)^3} \right) = \mathcal{O}\left( \frac{q_{\max}^2 |\mathcal{S}||\mathcal{A}|}{\epsilon^2(1-\gamma)^3} \log \frac{q_{\max}^2 |\mathcal{S}|^2 |\mathcal{A}|^2}{\delta\epsilon^2(1-\gamma)^3} \right). \tag{55}$$

It can be easily proved that among all the conditions $T_1$ has to satisfy the most restrictive is the one imposed by Corollary B.1. □

## C  Additional Experimental Results

In this section, we provide the experimental setup we adopted and some additional results we did not include in the main paper.

### C.1  Tabular RL

#### C.1.1  Experimental Setup

We train each agent for 100 episodes, the length of each episode is domain dependent. During the training periods we collect the rewards and use them to calculate the *online scores*. After each training period, we turn off exploration and evaluate the greedy policies learned by the agent. The length of the evaluation episodes is the same as the training episodes. We use the rewards collected during evaluation to calculate the *offline* scores. We perform this process in each domain, for each algorithm considered and show the mean scores of 10 runs with 95% c.i.. We calculate the undiscounted scores, even though we use a discount factor $\gamma = 0.99$ in each domain during learning.

For our particle algorithms, we initialize the particles equally spaced in an interval $[q_{\min}, q_{\max}]$, for each state action-pair. For the Gaussian model we initialized $\mu_0(s, a) = (q_{\max} + q_{\min})/2$ and $\sigma_0(s, a) = (q_{\max} - q_{\min})/\sqrt{12}$. The range of this interval is problem dependent and we see these hyperparameters as a way to incorporate prior knowledge about the domain. We consider Bootstrapped Q-learning with two policy models, the Bootstrapped policy defined in [30] and the posterior sampling policy. We initialize the Q-tables with values drawn from a Gaussian distribution with parameters $\mu = \frac{q_{\min}+q_{\max}}{2}, \sigma = q_{\max} - q_{\min}$. Furthermore, we consider Q-learning algorithm with $\epsilon$-greedy and Boltzmann exploration. In both Q-learning versions, the Q-table is initialized to 0. We compare our results with Delayed Q-learning [41], a model-free PAC-MDP algorithm. In each of the problems considered we tuned the $m$ parameter, the number of visits necessary to attempt an update for each state-action pair, to find the one that yields better results. We did not employ the theoretical values being too much conservative.

For all algorithms, we use an *exponentially decaying* learning rate given by:

$$\alpha_t(s, a) = \frac{b}{t(s, a)^a}, \tag{56}$$

where $t(s, a)$ is the visit count for state-action pair $(s, a)$, $b$ is the initial value which we set to 1 and $a$ is the *decay exponent*. We cross validated the value of $a$, which was set to $a = 0.2$, for all our experiments.

For the Q-learning algorithms, we had to chose also the schedules for $\epsilon$ and $\beta$, for $\epsilon$-greedy and Boltzmann exploration respectively. For $\epsilon$ we used an exponentially decaying schedule as in (56) with $b = 1$ and $a = 0.5$ whereas for the Boltzmann policy we used an exponentially decaying $\beta$ with initial value, $b = 1.5 q_{\max}$ and decay exponent, $a = 0.5$.

Here we show the results on more domains: Taxi, Chain and Loop domain from [15], River Swim and Six Arms from [42] and Knight Quest from [19].

## C.1.2 Comparison of WQL with State of the Art Algorithms

In Figure 4, we show a comparison between the best version of WQL, using the two models to approximate posteriors, and the considered RL algorithms.

Figure 4: Comparison of G-WQL, P-WQL, QL, BQL and Delayed QL. 10 runs, 95% c.i.

### C.1.3 Results of WQL algorithm

Figure 5 provides a full empirical analysis of the different flavors of WQL algorithms in the domains we considered.

(a) Chain      (b) Loop

(c) Taxi      (d) River Swim

(e) Six Arms      (f) Knight Quest

G-WQL-PX-PE    G-WQL-OX-ME    P-WQL-PS-OE
G-WQL-PX-ME    G-WQL-OX-OE    P-WQL-OX-PE
G-WQL-PX-OE    P-WQL-PS-PE    P-WQL-OX-ME
G-WQL-OX-PE    P-WQL-PS-ME    P-WQL-OX-OE

Figure 5: Results of different variations of WQL algorithm. 10 runs, 95% c.i.

### C.1.4 Effect of initialization in Particle WQL

We analyzed the effect of the initialization of the prior distributions in the particle case. We argue that the particle algorithm performs better when used with particles equally spaced in a given interval $[q_{\min}, q_{\max}]$. To show this we added noise to these equally spaced particles and ran the learning algorithm in the same domain. Figure 6 shows our results as a function of $\alpha$, denoting how spaced are the particles between each other. More specifically, $\alpha = 0$ means the particles were drawn uniformly random in the interval $[q_{\min}, q_{\max}]$ and $\alpha = 1$ means the particles are equally spaced in this interval. Any value in between is a combination of the two. For each $\alpha$, we averaged the learning curves of the agent. It seems clear to us that using equally spaced particles to represent the prior yields better results.

Figure 6: Effects of the initialization of the particles in Particle WQL. 10 runs.

### C.2 Deep RL

#### C.2.1 Experimental Setup

We test the algorithms using the Arcade Learning Environment (ALE). Each step of the agent corresponds to four steps of the emulator, where the same action is repeated. The reward values observed by the agents are clipped between -1 and 1 for stability. We evaluate our agents and report performance based upon the raw scores and not the discounted scores. As it is common in literature, we do not show the online performance of the agent during training. We show the scores collected, when exploiting the greedy policies derived from the Q-function after each training period.

The *convolutional* part of the network used is identical to the one used in [30]. The input to the network is $4 \times 84 \times 84$ tensor with a rescaled, grayscale version of the last four observations. The

first convolutional layer has 32 filters of size 8 with a stride of 4. The second layer has 64 filters of size 4 with stride 2. The last layer has 64 filters of size 3. We split the network beyond the final layer into $M = 10$ distinct heads, each one is fully connected and identical to the network in [30]. This consists of a fully connected layer to 512 units followed by another fully connected layer to the Q-Values for each action. The fully connected layers all use *Rectified Linear Units* (ReLU) as a *non-linearity*. We trained the networks with *RMSProp* optimizer. The discount was set to $\gamma = 0.99$, the number of steps between target updates was set to $\tau = 10000$ steps. The agents were evaluated every 1M frames.

The *experience replay* contains the 1M most recent transitions. We update the network every 4 steps by randomly sampling a minibatch of 32 transitions from the replay buffer to use the exact same minibatch schedule as Bootstrapped DQN.

### C.2.2 Network Initialization

An important problem we had to deal with was how to initialize the heads of the deep network. In the tabular case, we initialized the particles equally spaced in the interval $[q_{min}, q_{max}]$. We found it is not equally simple to extend this in the deep RL setting. We initialized the networks heads near the same interval by setting the bias of the last layer to the desired values.