[Reviews · NeurIPS 2019]

Reviewer 1



This paper proposes a mechanism for maintaining distributions over Q-values (called Q-posteriors) by defining the value function (the V-posterior) to be a Wasserstein barycenter of Q-posteriors and defining the TD update to be a Wasserstein barycenter of the current Q-posterior with an estimated posterior based on the value function. These distributions are intended to represent uncertainty about the Q-function and they enable more nuanced definitions of the "optimal" (w.r.t. actions) Q-value, for computing the V-posterior during TD learning, as well as for defining an exploration policy. Contributions seem to be: 1. A means of propagating uncertainty about Q-values via Wasserstein barycenters (Equations 2 & 3). 2. A proof that a modified version of the proposed algorithm is PAC-MDP in the average loss setting (Theorems 5.1 and 5.2). Empirical results: 1. Promising returns on four online RL tasks over tabular domains. 2. Favorable returns on the Asterix game, as compared to the Double DQN baseline. Strengths: 1. The paper is fairly clearly written and easy enough to understand. 2. The idea of propagating uncertainty via Wasserstein barycenters is interesting and suggests several concrete realizations. 3. Preliminary empirical results shows that propagating uncertainty can positively impact performance. Weaknesses: 1. The distinction between modeling uncertainty about the Q-values and modeling stochasticity of the reward (lines 119-121) makes some sense philosophically but the text should make clearer the practical distinction between this and distributional reinforcement learning. 2. It is not explained (Section 5) why the modifications made in Definition 5.1 aren't important in practice. 3. The Atari game result (Section 7.2) is limited to a single game and a single baseline. It is very hard to interpret this. Less major weaknesses: 1. The main text should make it more clear that there are additional experiments in the supplement (and preferably summarize their results). Questions: 1. You define a modified TD learning algorithm in Definition 5.1, for the purposes of theoretical analysis. Why should we use the original proposal (Algorithm 1) over this modified learning algorithm in practice? 2. Does this idea of propagating uncertainty not naturally combine with that of distributional RL, in that stochasticity of the reward might contribute to uncertainty about the Q-value? Typos, etc.: * Line 124, "... when experimenting a transition ..." ---- UPDATE: After reading the rebuttal, I have raised my score. I appreciate that the authors have included additional experiments and have explained further the difference between Definition 5.1 and the algorithm used in practice, as well as the distinction between the current work and distributional RL. I hope that all three of these additions will make their way into the final paper.

Reviewer 2



Thanks for the rebuttal and the extra results on two more atari games. I agree that this should be considered a theory paper and I'm keeping my score as is as I still think it's appropriate. ******************************************** The main idea of the paper is to construct a new TD update rule for the posterior over state-action values based on the notion of Wasserstein barycenter. This is an original idea which requires some time to digest. Based on this notion of TD learning, the authors derive a Q-learning algorithm which can take advantage of the estimated posterior for guiding exploration. They prove the efficiency of this algorithm in a specific tabular PAC-MDP setting, and they also attempt to scale the algorithm to situations where function approximation is necessary. The paper is very clearly written. Further comments: 1. The explanation of PDQN is not very detailed. There should be a separate algorithm box for PDQN. For example, the update rules considered in the paper are "derived" from gradient descent update rules and are directly tied to the notion of Wasserstein barycenter. DQN uses RMSProp update rule and so the connection to the theoretical results and definitions is unclear. 2. Why not show results for other Atari games? Asterix doesn't seem like the natural first choice for evaluating an algorithm. It seems that there are some extra hyperparameters e.g. related to the choice of the prior. The extra flexibility that stems from tuning these parameters could improve performance due to reasons unrelated to having an access to a good posterior.

Reviewer 3



-------------------------------------- Post rebuttal ---------------------------------------- I thank authors for providing the rebuttal, specifically for clarifying the relevance of OT to the problem considered and for agreeing to extend the discussion of OT use-cases in RL. I am increasing my score. ---------------------------------------------------------------------------------------------- The paper is rather clearly written. Detailed supplementary material was helpful to clarify several questions I had while reading the main text. It seems that Proposition A.3 requires a bit of a revision for the particle case. For a 1d Wasserstein barycenter to be an average, the particles should be ordered as in the Proposition A.1, but nothing about ordering is stated in A.3. One of the most intriguing properties of OT is its ability to take into account the geometry of the support of the distributions. In this paper OT is used for the value-function posteriors, which have 1d support without any apparent geometric properties. This also leads to quite trivial barycenters. My question is what is the reason to use Wasserstein barycenters over some other way of averaging distributions? Perhaps a more promising direction to incorporate OT into RL is to consider distributions over the state space. In many RL applications state space has interesting geometric properties which seem to be ignored by standard RL algorithms. I am wondering what authors think about such perspective and the premise of OT in RL in general.

[Author Response · NeurIPS 2019]

We thank the reviewers for their insightful comments and suggestions. We hope that the rebuttal will clarify the issues.

**Reviewer #1**

1. (*Modified Algorithm of Definition 5.1*) The update rules of Definition 5.1 (leading to Algorithm 2 in the suppl. material) reduce to Definition 3.2 (leading to Algorithm 1) when $\mathcal{Q}_b = 0$ a.s. or $\beta_t = 0$. $\mathcal{Q}_b$ is used in the analysis as an additional source of uncertainty necessary to prove, together with a suitable choice of $\alpha_t$ and $\beta_t$, the PAC-MDP in the average loss for *any* tabular MDP. There are two reasons why Algorithm 1 should be preferred in practice. First, Algorithm 2 cannot be extended to continuous MDPs, as $\alpha_t$ and $\beta_t$ are defined in terms of number of visits $n(s, a)$ (Equation (18) in the suppl. material), which can only be computed for finite MDPs. Second, as many provably efficient RL algorithms (e.g., MBIE or Delayed-QL), Algorithm 2 is extremely conservative, leading to very slow convergence. This is why most provably efficient RL algorithms, when used in practice, are run with non-theoretical values of hyperparameters (see Figures 2 and 3 of [2] for MBIE). Algorithm 1 can be seen as a "practical" version of Algorithm 2 in which $\alpha_t$ is treated as a normal hyper-parameter and $\beta_t = 0$. We will clarify this point in the final version.

2. (*Comparison with Distributional RL (DRL)*) For space reasons, we condensed the discussion in lines 119-121. While DRL models the distribution of the *return*, our WQL models the distribution of the Q-function estimate, which is defined as the *sample mean* of the *returns*. The two distributions are clearly related and both depend on the stochasticity of the reward and of the transition model. The main difference is that DRL quantifies the intrinsic stochasticity of the return, while in WQL the stochasticity refers to the uncertainty on the Q-function estimate which reduces as the number of updates increases, being a sample mean. We will reserve more space for this comparison in the final version.

3. (*Experiments*): we compared PDQN with two baselines: the standard Double DQN (DDQN) and Bootstrapped DQN (BDQN), meant to enforce exploration. See **General Note on Experiments** for details.

**Reviewer #2**

1. (*PDQN*) We are aware that the description of PDQN is synthetic, due to space constraints. We believe that the application to deep-RL should not be considered the main focus of the paper. As stated in **General Note on Experiments** our goal with PDQN is purely illustrative, showing the ability of WQL to be applied to continuous domains. More details about PDQN and the experimental setting are reported in Appendix C.2. We will insert the pseudocode of the PDQN algorithm in the final version. The update rules used in the case of a particle model for the Q-posteriors are reported in Table 1. The learning rate $\alpha_t$ can be set using for instance RMSProp. We stress that our theoretical findings (Section 5) are limited to the tabular case with no function approximation and, thus, are not applicable to PDQN. Extending the theory to function approximation is an appealing future research direction.

2. (*Experiments*) In PDQN the prior is the spread of the particles (see Appendix C.2.2). It determines the amount of exploration (similarly to BDQN) and how posteriors are built/updated. See also **General Note on Experiments**.

**Reviewer #3**

1. (*Proposition A.3*) We assumed that the particles are ordered as in Proposition A.1, we will state the assumption.

2. (*About the choice of Wasserstein*) The main reason why we chose the Wasserstein metric over other distributional distances (e.g., $\alpha$-divergences) is that the Wasserstein distances are able to deal with deterministic distributions ($\alpha$-divergences degenerate to infinity). This feature is important for us, as the Q-posteriors model a sample mean and, thus, their variances reduce as the number of samples increases, moving towards a deterministic distribution. This case is discussed in Proposition 3.1, in which the geometry of the support ($\mathbb{R}$) becomes essential in order to average deterministic distributions and recover the standard TD solution. This would not be possible if we employed $\alpha$-divergences. Exploiting OT to account for the geometry of the state space is an appealing idea, although rather different from what we propose in our paper. In this regard, there are several works that exploit Lipschitz assumptions to incorporate the geometry of the MDP also using the Wasserstein (or Kantorovich) distances (e.g., [1]). We will explain, in the final version, the geometric motivations behind the choice of the Wasserstein distance and we will discuss in Section 7 the use of OT in RL.

**General Note on Experiments** We are aware that our experiments on Atari games are very limited. However, we believe that the focus of the paper is not the proposal of an effective deep-RL algorithm, but an algorithm that uses modern notions of Wasserstein barycenters, endowed with strong theoretical guarantees (PAC-MDP in average loss), which can be easily extended to function approximation, unlike other provably-efficient algorithms (e.g., MBIE, Delayed-QL). The choice of the Asterix game derives from the fact that, to see

the advantages of PDQN, we need an environment in which exploration is essential. During the rebuttal period we run experiments on new environments: Breakout and Enduro (see figure). Unfortunately, due to our limited infrastructure, we could only show the average of 3 runs. Since in Breakout exploration is not an issue, the performance of PDQN is similar to BDQN but better than DDQN. However, in Enduro we can appreciate that PDQN learns substantially faster w.r.t. BDQN and DDQN, as exploration becomes relevant. We stress that all 3 algorithms use the same hyper-parameters (network, learning rate, replay buffer), which were not tuned. We will include these experiments in the final version.

[1] E. Rachelson and M. G. Lagoudakis. On the locality of action domination in sequential decision making. *ISAIM*, 2010.

[2] A. L. Strehl and M. L. Littman. An analysis of model-based interval estimation for markov decision processes. *Journal of Computer and System Sciences*, 74(8):1309–1331, 2008.


[Meta-Review · NeurIPS 2019]

This paper was enthusiastically received by the reviewers --- congratulations! In your revision, please address comments and requests in the reviews.